# Enhancing the dark side: asymmetric gain of cone photoreceptors underpins their discrimination of visual scenes based on skewness

Matthew Yedutenko[1] , Marcus H. C. Howlett[1] and Maarten Kamermans[1,2]

[1]*Retinal Signal Processing Laboratory, Netherlands Institute for Neuroscience, Amsterdam, The Netherlands*
[2]*Department of Biomedical Physics and Biomedical Optics, Amsterdam University Medical CenterUniversity of Amsterdam, Amsterdam, The Netherlands*

Edited by: Ian Forsythe and William Taylor

The peer review history is available in the Supporting Information section of this article (https://doi.org/10.1113/JP282152#support-information-section).

**Matthew Yedutenko** received his BSc in applied mathematics and physics and MSc in molecular physiology and biophysics in Moscow Institute of Physics and Technology. His master's thesis was focused on the neuronal calcium sensors and was conducted in the laboratory of Professor Pavel Belan at the National Academy of Sciences of Ukraine (NASU) Bogomoletz Insitute of Physiology in Kyiv, Ukraine. Currently, Mr Yedutenko is a PhD student in Professor Maarten Kamerman's group at the Netherlands Institute for Neuroscience in Amsterdam, where he is investigating the functional properties of cone photoreceptors and the implications of these properties for downstream visual processing. His research is funded by the European training network 'SwitchBoard' under the Marie-Sklodowska Curie Grant Actions.

This article was first published as a preprint. Yedutenko M, Howlett MHC, Kamermans M. 2021. Enhancing the dark side: Asymmetric gain of cone photoreceptors underpins discrimination of visual scenes based on their skewness. bioRxiv. https://doi.org/10.1101/2021.04.29.441966.

The Journal of Physiology

**Abstract** Psychophysical data indicate that humans can discriminate visual scenes based on their skewness, i.e. the ratio of dark and bright patches within a visual scene. It has also been shown that at a phenomenological level this skew discrimination is described by the so-called blackshot mechanism, which accentuates strong negative contrasts within a scene. Here, we present a set of observations suggesting that the underlying computation might start as early as the cone photo-transduction cascade, whose gain is higher for strong negative contrasts than for strong positive contrasts. We recorded from goldfish cone photoreceptors and found that the asymmetry in the phototransduction gain leads to responses with larger amplitudes when using negatively rather than positively skewed light stimuli. This asymmetry in amplitude was present in the cone photocurrent, voltage response and synaptic output. Given that the properties of the phototransduction cascade are universal across vertebrates, it is possible that the mechanism shown here gives rise to a general ability to discriminate between scenes based only on their skewness, which psychophysical studies have shown humans can do. Thus, our data suggest the importance of non-linearity of the early photoreceptor for perception. Additionally, we found that stimulus skewness leads to a subtle change in photoreceptor kinetics. For negatively skewed stimuli, the impulse response functions of the cone peak later than for positively skewed stimuli. However, stimulus skewness does not affect the overall integration time of the cone.

(Received 12 July 2021; accepted after revision 11 November 2021; first published online 15 November 2021)
**Corresponding author** Maarten Kamermans: Retinal Signal Processing Laboratory, Netherlands Institute for Neuroscience, Meibergdreef 47. Amsterdam, 1105 BA, The Netherlands. Email: m.kamermans@nin.knaw.nl

**Abstract figure legend** From left to right. An example of a texture with stripes of alternating skewness employed in psychophysical studies. An array of cone photoreceptors, each of which possess a non-linear gain function (middle-top). The cone's non-linear gain leads to responses with larger amplitudes (right-top) when negatively skewed light stimuli are used than when positively skewed light stimuli are used (right-bottom).

## Key points

- Humans can discriminate visual scenes based on skewness, i.e. the relative prevalence of bright and dark patches within a scene.
- Here, we show that negatively skewed time-series stimuli induce larger responses in goldfish cone photoreceptors than comparable positively skewed stimuli.
- This response asymmetry originates from within the phototransduction cascade, where gain is higher for strong negative contrasts (dark patches) than for strong positive contrasts (bright patches).
- Unlike the implicit assumption often contained within models of downstream visual neurons, our data show that cone photoreceptors do not simply relay linearly filtered versions of visual stimuli to downstream circuitry, but that they also emphasize specific stimulus features.
- Given that the phototransduction cascade properties among vertebrate retinas are mostly universal, our data imply that the skew discrimination by human subjects reported in psychophysical studies might stem from the asymmetric gain function of the phototransduction cascade.

## Introduction

Psychophysical studies show that humans are sensitive to the ratio of negative (intensity lower than the mean) and positive (intensity higher than the mean) patches of contrast in visual scenes (Chubb *et al.* 1994, 2004; Graham *et al.* 2016). This ratio is described by the parameter known as skewness. Visual stimuli are called positively

skewed if there is a predominance of negative contrasts with some infrequent patches of high positive contrast and negatively skewed when the situation is reversed. Figure 1 illustrates one's ability to discriminate visual scenes based on skewness by mimicking an experiment performed by Chubb *et al.* (1994, 2004). The textures were randomly drawn from two distributions equal in every aspect but

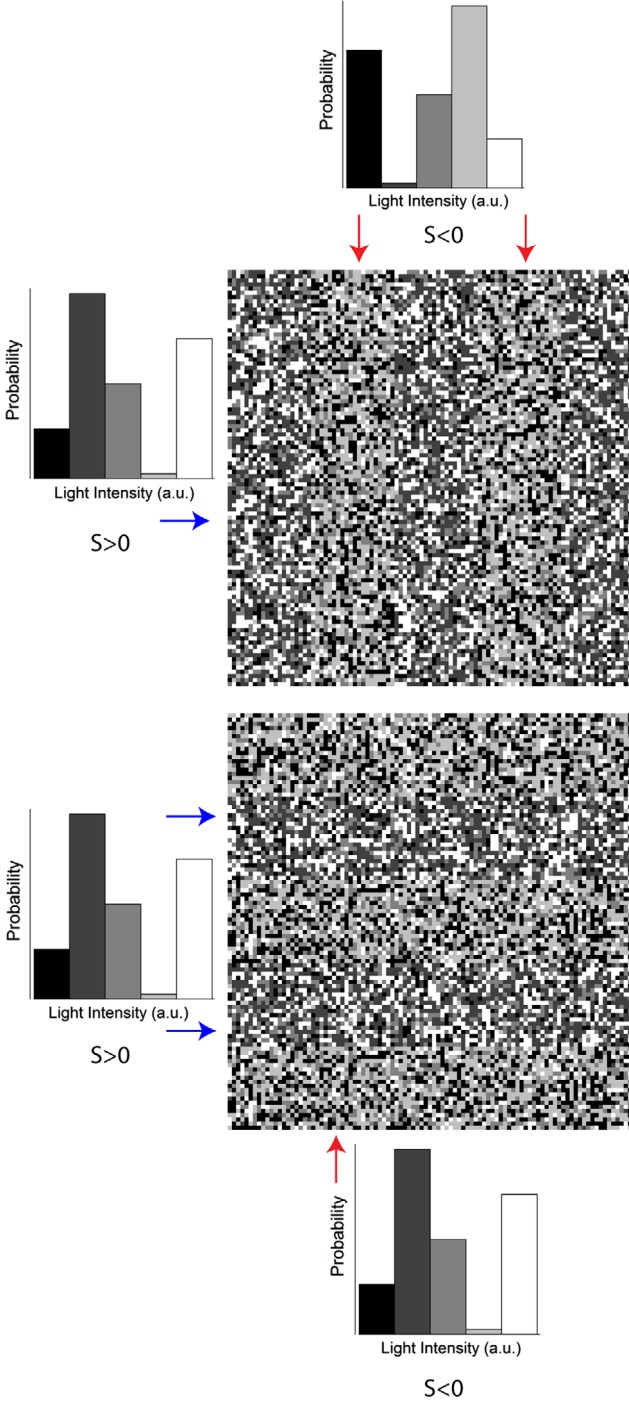

**Figure 1. The discrimination of skewed stimuli**
An example of textures used by Chubb *et al*. (1994) to probe psychophysically the ability of humans to discriminate textures based on skewness. The textures consisted of the either vertical (upper) or horizontal (lower) bars of alternating skewness. The participant was asked to judge the orientation of the bars. Arrows indicate the probability distribution from which the corresponding bar was drawn. The bars were drawn from the probability distributions following the approach described by Bonin *et al*. (2006) and differed only in terms of skewness (±0.4). [Colour figure can be viewed at wileyonlinelibrary.com]

skewness (Bonin *et al*. 2006). Yet, one can appreciate the clear difference between negatively skewed images (right upper) and positively skewed images (remaining panels). Chubb *et al*. (1994, 2004) showed that on a phenomenological level, the sensitivity to skewness can be described by the so-called blackshot mechanism. This blackshot mechanism does not react to skewness *per se*, rather its sensitivity to strong negative contrasts is simply much higher than to strong positive contrasts, hence it effectively reports the fraction of strong negative contrasts within the scene.

What are the neuronal correlates of the blackshot mechanism? Studies using salamander retinal ganglion cells (Tkačik *et al*. 2014) and cat lateral geniculate nucleus (LGN) neurons (Bonin *et al*. 2006) did not report any differences in response associated with changes in stimulus skewness. Therefore, both studies concluded that the discrimination between skewed stimuli occurs in the visual cortex. In contrast, it is well established that the gain of the retinal photoreceptor is asymmetric; given an equal stimulus magnitude, the response amplitude to a strong (>0.4 Weber unit) negative contrast step is greater than it is to a strong positive contrast step (Laughlin, 1981; van Hateren, 2005; Endeman & Kamermans, 2010; Baden *et al*. 2013; Angueyra *et al*. 2021). Furthermore, this responses asymmetry is observed throughout the post-receptor retinal stages (Lee *et al*. 2003; Zaghloul *et al*. 2003), the LGN (Kremkow *et al*. 2014) and the primary visual cortex (Zemon *et al*. 1988; Jin *et al*. 2008; Yeh *et al*. 2009; Kremkow *et al*. 2014). The asymmetric processing of positive and negative contrasts should lead to different response amplitudes to negatively and positively skewed stimuli and thus might underpin the discrimination of skewed stimuli.

A possible reason why the differences in responses to skewed stimuli were not found in retinal ganglion cell (Tkačik *et al*. 2014) and LGN (Bonin *et al*. 2006) studies was the power spectra of the stimuli used. In both cases, the researchers used band-limited white noise, whereby large proportions of the signal power are outside the photoreceptor frequency bandwidth. Thus, in both studies, temporal filtering discarded a significant portion of the signal, reducing the skewness and amplitude of the 'effective' light stimuli available to drive the photoreceptor non-linear gain. Consequently, Bonin *et al*. (2006) and Tkačik *et al*. (2014) might not have found significant differences in the processing of skewed stimuli because (a) their stimuli hardly differed in terms of 'effective' skewness, and (b) the 'effective' amplitudes of the stimuli used were too low to drive cone photoreceptors outside their linear response range.

Here, we stimulated goldfish cones with sets of skewed stimuli with bandwidths similar to those of the goldfish cones and found that responses to negatively skewed

stimuli indeed have higher amplitude than responses to positively skewed stimuli.

## Methods

### Ethical approval

All animal experiments were conducted under the responsibility of the ethical committee of the Royal Netherlands Academy of Arts and Sciences (KNAW), acting in accordance with the European Directive 2010/63 of the European Parliament and of the Council, under license number AVD-801002016517, issued by the Central Comity Animal Experiments of The Netherlands. In addition, every possible measure was taken to reduce any potential suffering and the number of animals used. For all experiments, retinas from 10- to 15-cm-long adult (6–18 months) goldfish (*Carassius auratus*) of either sex were used. The fish were supplied by Gommers-Ducheine BV (Ghent, The Netherlands) and held in tanks with a water temperature of 15°C in a 12 h–12 h light–dark cycle and were fed six times per week.

### Recording procedures

On the day of the experiment, goldfish were first dark adapted for 5–10 min, killed by decapitation and their eyes enucleated. Retinas were isolated under dim red illumination, then placed photoreceptor side up in a recording chamber (300 $\mu$l; model RC-26G, Warner Instruments) mounted on a Nikon Eclipse 600FN microscope. The preparation was viewed on an LCD monitor by means of a ×60 water-immersion objective (NA 1.0; Nikon), a CCD camera and infrared ($\lambda > 800$ nm; Kodak wratten filter 87c, USA) differential interference contrast optics. The tissue was continuously superfused with oxygenated Ringer solution at room temperature (20°C). The composition of the Ringer solution was as follows (mM): 102.0 NaCl, 2.6 KCl, 1.0 MgCl$_2$, 1.0 CaCl$_2$, 28.0 NaHCO$_3$ and 5.0 glucose, continuously gassed with 2.5% CO$_2$ and 97.5% O$_2$ to yield a pH of 7.8 (osmolarity 245–255 mosmol l$^{-1}$). For calcium current ($I_{Ca}$) measurements, 5 mM of NaCl was replaced with 5 mM of CsCl, and 100 $\mu$M of niflumic acid was added.

Measurements from goldfish cones were performed in current-clamp (voltage response) and voltage-clamp (photo- and calcium current) configurations. Patch pipettes (resistance 7–8 M$\Omega$, PG-150T-10; Harvard Apparatus, Holliston, MA, USA) were pulled with a Brown Flaming Puller (model P-87; Sutter Instruments, Novato, CA, USA). The patch pipette solution contained (mM): 96 potassium gluconate, 10 KCl, 1 MgCl$_2$, 0.1 CaCl$_2$, 5 EGTA, 5 HEPES, 5 ATP-K$_2$, 1 GTP-Na$_3$, 0.1 cGMP-Na, 20 phosphocreatine-Na$_2$ and 50 units ml$^{-1}$ creatine phosphokinase, adjusted with KOH to pH 7.27–7.3 (osmolarity 265–275 mosmol l$^{-1}$). The chloride equilibrium potential ($E_{Cl}$) was $-55$ mV except when $I_{Ca}$ was studied. Here, $E_{Cl}$ was set at $-41$ mV by changing the concentrations of potassium gluconate and KCl to 87 and 19 mM, respectively. All chemicals were supplied by Sigma-Aldrich (Zwijndrecht, The Netherlands), except for NaCl (Merck Millipore, Amsterdam, The Netherlands).

Filled patch pipettes were mounted on a MP-85 Huxley/Wall-type manual micromanipulator (Sutter Instrument) and connected to a HEKA EPC-10 Dual Patch Clamp amplifier (HEKA Elektronik, Lambrecht, Germany). After obtaining a whole-cell configuration, cones were first classified spectrally based on their response amplitudes to long-, mid- or short-peak wavelength light flashes (see next subsection). Subsequent light stimuli were generated using only the light source that induced that largest light response during the spectral classification stage. Only cells with stable and fast light responses were stimulated with light skewed stimuli. Data were recorded at a sample rate of 1 kHz using the Patchmaster software package (HEKA Elektronik).

In total, we recorded from 14 cones in eight animals in voltage-clamp mode (light responses, eight cones in seven animals; $I_{Ca}$ measurements, six cones in one animal) and from 16 cones in 14 animals in current-clamp mode (all light responses).

For the skew stimulus set 1 (see next subsection) conditions, in voltage clamp we recorded from one short-wavelength-sensitive cone (S-cones), six middle-wavelength-sensitive cones (M-cones) and one long-wavelength-sensitive cone (L-cones); and in the current-clamp mode from two S-cones, six M-cones and one L-cone. For the skew stimulus set 2 conditions, we recorded two S-cones and five M-cones in current-clamp mode only. No spectral classification of the cones was done in the experiments concerning measurements of the $I_{Ca}$.

### Light stimuli

The light stimulator was a custom-built LED stimulator with a three-wavelength high-intensity LED (Atlas, Lamina Ceramics, Westhampton, NJ, USA). The peak wavelengths were 465, 525 and 624 nm. Bandwidth was < 25 nm. Linearity was ensured by an optical feedback loop. The output of the LED stimulator was coupled to the microscope via an optical fibre and focused on cone outer segments though a ×60 water-immersion objective. The mean light intensity of all stimuli was $1.2 \times 10^4$ photons $\mu$m$^{-2}$ s$^{-1}$, which is in the photopic level for goldfish (Malchow & Yazulla, 1986). Stimuli were presented at 1 kHz.

**Skew stimulus set 1.** Skewed stimuli were based on the natural time series of chromatic intensities (NTSCI) from the Van Hateren library (Van Hateren *et al.* 2002). Given that psychophysical studies have shown that visual scene discrimination occurs when intensities within the spatial domain are skewed (Fig. 1), one may wonder how appropriate it is to study the phenomena with time series of intensities. However, we argue that such an approach is correct for the following two reasons. Firstly, saccadic eye movements convert spatial stimuli into time series. Hence, the photoreceptors of subjects in psychophysical studies 'perceive' the presented skewed textures as time series of intensities. Secondly, direct light response of a photoreceptor depends only on the light intensities falling upon its outer segment and not on the spatial structure of the stimulus. Hence, one can reproduce the response of an array of cones to the presentation of a single spatial texture by recording the response of a single cone to a collection of time series of intensities that correspond to the differing retinal locations.

The NTSCI power spectrum is typical of that of 'natural stimuli' in that power declines as a function of frequency (Van Hateren, 1997; Van Hateren *et al.* 2002; Frazor & Geisler, 2006). As a result of the predominance of lower frequencies, most of the light-intensity changes throughout the NTSCI occur over time scales accessible to goldfish cones, and previously, the NTSCI has been used to unlock several non-linear performance features of cones (Endeman & Kamermans, 2010; Howlett *et al.* 2017). To ensure that all aspects other than skewness remained equal, we first picked short stretches from the NTSCI that were positively skewed, then simply flipped these around the mean to generate negatively skewed stimuli.

Van Hateren's NTSCI is composed of intensity profiles for three distinct chromatic channels that, although similar, are not equal. Here, we used only the 'red' channel intensity profile of the NTSCI to generate our skewed stimuli. These stimuli then drove the LED with the peak wavelength best matched to the spectral type of the recorded cone. The LED intensities were also adjusted such that each cone type received an equal number of quanta. In this way, we ensured that we delivered the same stimuli to cones regardless of their spectral type.

To generate stimulus set 1, we divided the NTSCI (Van Hateren *et al.* 2002) into 1-s-long stretches. From each stretch, we subtracted its minimum value, adjusted the mean light intensities to be equal and selected stretches with similar power spectra, root mean square (r.m.s.) and median contrasts (between 0.23 and 0.25). The r.m.s. and median contrast for each stretch was calculated, respectively, as the ratio between the standard deviation and mean of the stretch, and the ratio between its deviation from the median and its median. To ensure an absolute similarity between positively and negatively skewed stimuli, we selected only stretches where the maximum value was not larger than two times the mean. Next, we chose stretches with skewness values of 0.9, 1.6 and 2.2. The skewness was calculated with Eq. (1):

$$S = \left\langle \frac{(I - I_{\text{mean}})^3}{N\sigma^3} \right\rangle \tag{1}$$

where $N$ is the number of elements in the stretch, $I$ corresponds to the light intensity of an element, $I_{\text{mean}}$ and $\sigma$ are the mean and standard deviation within the stretch, and the angle brackets denote averaging over the period.

We narrowed our selection further to three stretches, all with similar power spectra (data not shown). Power spectra were calculated by Welch's averaged periodogram method (Welch, 1967). No window function was used; the length of the Fourier transform was same as the length of each corresponding data sequence. The total stimulus power was calculated as the integral under power spectra, and the differences in the total stimulus power were no more than 10%. Finally, an additional pink noise stimulus with zero skew and similar power spectra was added to the set. In total, skew stimulus set 1 consisted of seven 1 s stimuli.

**Skew stimulus set 2.** This stimulus set consisted of three 4-s-long stretches with a skewness of 2.2, 0 and −2.2. They were generated in the same way as skew stimulus set 1, but with one additional condition, i.e. the degree of skewness delivered by the stimulus remained unchanged by the temporal filtering of the cone. This was ensured by first convolving the NTSCI stretch with the mean photo-current impulse response function (see Data Analysis) obtained from responses to skew stimulus set 1. The skew of the convolution product, representing the 'effective' stimulus, was then compared with the skew of the original stimulus. This was also confirmed by determining the effective skewness after convolving the stimuli with the impulse response function of each cone measured in skew stimulus set 2 conditions.

### Calcium current isolation

To measure $I_{\text{Ca}}$, we used the mean voltage response (seven cells, 69 repeats in total) of cone photoreceptors to stimulus set 2 as the command voltages for the voltage-clamp experiments.

To isolate $I_{\text{Ca}}$, we followed the approach described by Fahrenfort *et al.* (1999). Briefly, to eliminate the calcium-dependent chloride current, $E_{\text{Cl}}$ was set at −41 mV, and 100 $\mu$M niflumic acid was added to the Ringer solution; delayed rectifying and hyperpolarization-activated potassium currents were blocked by replacing 5 mM NaCl in the Ringer solution with 5 mM CsCl; light-activated conductances were saturated by a 20 $\mu$m spot of bright white light focused

on the cone outer segment; linear leak currents were removed by subtraction. The leak current was estimated by clamping cones at $-70$ mV, stepping to potentials between $-100$ and 20 mV in 5 mV steps for 100 ms, calculating the mean current between 20 and 60 ms after the step onset, then determining the linear fit of the current–voltage relationship between $-100$ and $-60$ mV (Vroman *et al.* 2014; Kamar *et al.* 2019).

## Data analysis

For each cell, the skewness of its mean response to each stimulus was determined using Eq. (1). In Figs 2, 3*D* and 8*A*, data were fitted using built-in Matlab least-square methods. All data analysis was performed in Matlab and Python.

Parallel cascade identification is the most rigorous method to describe the signal-processing properties of cone responses to naturalistic stimuli (Korenberg, 1991). However, for practical reasons our analysis focuses only on the estimation of the first parallel cascade, which is effectively a linear filter followed by a static non-linearity. Apart from the computational and descriptive simplicity, this approach is also justifiable because it describes cone responses accurately, accounting for >95% of the variance.

The linear temporal filtering properties of a cone were described by its impulse response function. Impulse response functions were estimated as the inverse Fourier transform of the ratio between the stimulus–response cross-power and stimulus–power spectra (Wiener, 1964; Kim & Rieke, 2001). The spectra were calculated using Welch's averaged periodogram method (Welch, 1967). Stimuli and responses were detrended, divided into 500-ms-long stretches with 50% overlap, and windowed with a Hamming function. The length of the Fourier transform was 1024 ms. For Figs 3–5 and 7, impulse response functions were averaged across the entire skew stimuli sets to avoid biases in their estimate associated with skewness of the individual stretches (Chichilnisky, 2001; Simoncelli *et al.* 2004; Bonin *et al.* 2006; Tkačik *et al.* 2014).

To estimate the 'effective' stimuli, we convolved the impulse response functions of the cone with the 'original' light stimuli. Skews of these 'effective' stimuli were calculated with Eq. (1). Discrepancies between the skewness of the 'effective' stimuli and the skewness of the responses of the cone were considered to be a result of non-linear cone properties.

For 'effective' Weber contrast steps (Figs 7 and 8*A*), light stimuli were first converted into Weber contrast steps (Fig. 7) with Eq. (2):

$$C = \frac{(I - I_{\mathrm{mean}})}{I_{\mathrm{mean}}} \qquad (2)$$

These Weber contrast steps (C) were then convolved with a mean impulse response function to obtain the 'effective' Weber contrast steps. The mean impulse response function used here was the averaged voltage

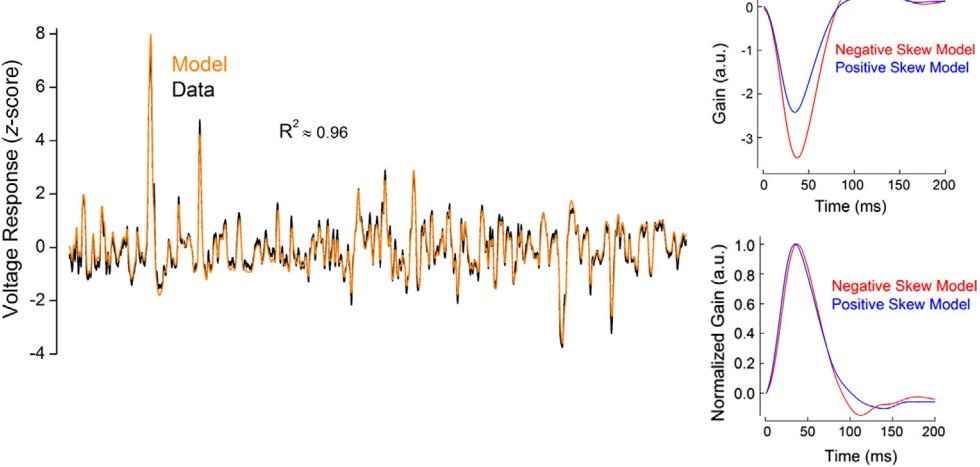

**Figure 2. Example of the performance of Van Hateren's model for goldfish cones**
Left panel, voltage responses as *z*-scores to skew stimulus set 2 for a representative recorded cone (black line) and for the simulated cone (orange line). The coefficient of determination ($R^2$) between these two traces was 0.96. Right panel, examples of the impulse response functions obtained from the simulated responses to negatively (red) and positively (blue) skewed stimuli. Similar to the impulse response functions estimated from the recorded responses (Fig. 8*B* and *C*), the simulated impulse response functions peaked 3 ms [8.5 (2.64)%, $n = 7$, $P = 0.00039$, Student's paired *t*-test] later for the negatively skewed stimulus compared with the positively skewed stimulus and showed no statistically significant difference in their full width at half-maximum [FWHM; $\Delta = 0.006$ (0.019)%, $n = 7$, $P = 0.447$, Student's paired *t*-test] or integration time [$\Delta = -2.6$ (2.53)%, $n = 7$ $P = 0.0512$, Student's paired *t*-test]. Parameters for the simulation are listed in the Table 1. [Colour figure can be viewed at wileyonlinelibrary.com]

response-derived impulse response function of all 16 cones measured in current clamp, which was subsequently scaled such that the integral under its curve yielded one (Fig. 7; Howlett *et al.* 2017).

For Figs 2 and 8, we calculated impulse response functions using responses to individual skewed stretches. In this instance, the impulse response functions estimated might differ from the 'true' impulse response function of the system because of the biases associated with the skewness of stimuli. Nevertheless, given that the positively and negatively skewed stimuli in skew stimulus set 2 differed only in terms of skewness, the comparison of corresponding impulse response functions still enables any potential differential effect of stimulus skewness on cone signal-processing properties to be assessed. From these data, the cone integration time was calculated as the integral of the initial polarization lobe of the impulse response function (Daly & Normann, 1985).

To estimate the non-linear gain function parameters of the cone, we fitted the relationship between all the mean cone voltage responses and the 'effective' Weber contrast steps (Fig. 8*A*; 19,000 data points) as the power function of input contrast (Van Hateren & Snippe, 2006) with Eq. (3):

$$\Delta V = a \times (C + b)^d + e \tag{3}$$

Here, $\Delta V$ denotes the voltage response, $C$ is the 'effective' Weber contrast, and $a$, $b$, $d$ and $e$ are fit parameters. At the biophysical level, $b$ corresponds to the baseline rate of phosphodiesterase activity (PDE), $d$ describes the interdependence between the PDE activity rate and the voltage response, $e$ is proportional to the baseline concentration of cyclic guanosine monophosphate (cGMP), and $a$ is a scaling factor (Van Hateren & Snippe, 2006). The quality of the fit was quantified with the adjusted coefficient of determination ($R^2$). The highest $R^2$ (0.95) was obtained using the following parameter values (value $\pm$ 95% confidence interval): $a = 0.05138 \pm 0.03506$, $b = 1.166 \pm 0.072$, $d = -0.1251 \pm 0.0959$ and $e = -0.05046 \pm 0.03552$. Our estimation of the power function ($d$) was close to that obtained by Van Hateren & Snippe (2006) in their theoretical study.

## Model

Photoreceptor responses were modelled in Matlab using Van Hateren's model of vertebrate photoreceptors (van Hateren & Snippe, 2007), which was shown to be remarkably precise in capturing the processing steps involved in generating the signal of a cone. Apart from activation of the hyperpolarization-activated current ($I_h$; Howlett *et al.* 2017; Kamermans *et al.* 2017), the model closely simulates all the biophysical processing steps of the cone, from the photon-initiated activation

of conopsins to the cGMP-regulated changes in the photocurrent, followed by generation of the voltage response. The model simulates cone photoreceptors as a cascade of low-pass filters, a static (instantaneous and memoryless) non-linearity and two divisive feedback loops (van Hateren, 2005; Van Hateren & Snippe, 2007). The low-pass filters correspond to the kinetics of the different biophysical processing steps. The non-linearity describes the inverse proportional dependence between light intensity and changes in the cGMP concentration. The first feedback loop describes the regulation of the rate of cGMP production by calcium influx through cGMP-gated channels. The second feedback loop corresponds to the regulation of the membrane voltage by voltage-sensitive channels in the cone inner segment. The non-linear gain of the cone (Fig. 8*A*) originates from the interplay between the hydrolysis of cGMP by PDE and the calcium-regulated (feedback loop) production of cGMP by guanylyl cyclase (GC).

We verified that Van Hateren's model could capture responses to skewed stimuli. For this, we fitted the model to the voltage responses of seven goldfish cones recorded under skew stimulus set 2 conditions. The model parameters were modulated within the ranges determined by Endeman & Kamermans (2010) and are shown in Table 1. On average, the correlation coefficient between modelled and recorded voltage responses was 0.97 (0.037), with a coefficient of determination of 0.95 (0.007) (Fig. 2). Moreover, the impulse response functions estimated from the simulated responses to positively and negatively skewed stimuli retained features of the impulse response functions derived from the recorded voltage responses. For example, for both recorded and simulated cone voltage responses, impulse response functions peaked 3 ms [8.5 (2.64)%, $P = 0.00013$, Student's paired *t*-test] later under the negatively skewed stimulus compared with the positively skewed conditions, but showed no statistically significant difference in their full width at half-maximum (FWHM) or in integration time (Fig. 2). Thus, Van Hateren's model reproduces cone responses to skewed stimuli accurately.

Next, we used Van Hateren's model to estimate the 'effective' stimuli in the salamander (Tkačik *et al.* 2014) and cat (Bonin *et al.* 2006) studies. To model salamander cones, we adjusted the parameters of Van Hateren's model such that the time course of the impulse response functions of simulated cones resembled those of salamander cones reported by Rieke (2001) and Baccus & Meister (2002). The exact simulation parameters are reported in Table 1. Likewise, to model cat cones Van Hateren's model parameters were adjusted such that the impulse response function time course resembled the estimates made by Donner & Hemila (1996). For the cat, exact parameters of the simulation are reported in the Table 1.

**Table 1. Parameters of Van Hateren's model used to simulate the responses of goldfish, salamander and cat cones**

| Parameter | Goldfish cones | Cat cones | Salamander cones |
|---|---|---|---|
| Lifetime of activated conopsin (ms) | 8–31 | 8 | 88 |
| Lifetime of activated transducin (ms) | 16–30 | 12 | 101 |
| Dark phosphodiesterase activity (ms$^{-1}$) | 0.003 | 0.0028 | 0.003 |
| Constant of the dependence of phosphodiesterase activity on transducin activation | 0.00004–0.00023 | 0.00016 | 0.0002 |
| Apparent Hill coefficient of CNG channels | 1 | 1 | 1 |
| Hill coefficient of guanylyl cyclase activation | 4 | 4 | 4 |
| Time constant of calcium extrusion (ms) | 12–28 | 9 | 24 |
| Guanylyl cyclase activation constant | 0.1 | 0.09 | 0.1 |
| Capacitive membrane time constant (ms) | 15 | 6 | 15 |
| Parameter of membrane non-linearity | 0.7–1.1 | 0.8 | 0.85 |
| Constant of membrane non-linearity | 0.03–0.07 | 0.07 | 0.085 |
| Time constant of membrane non-linearity (ms) | 300 | 120 | 300 |

For the goldfish, model parameters were obtained by fitting cone responses ($n = 7$) to skew stimulus set 2 while constraining the range of the parameters varied to within that determined by Endeman & Kamermans (2010). For the cat and salamander, parameters were adjusted such that the impulse response function time to peak of the simulated cone approximately matched that estimated, respectively, by Donner & Hemila (1996) and by Rieke (2001) and Baccus & Meister (2002).

'Light' stimuli mimicking those used by Bonin *et al.* (2006) and Tkačik *et al.* (2014) were used to study the responses of the modelled cat (Fig. 9) and salamander (Fig. 10) cones, respectively, to changes in skewness. The only difference was that, for illustrative ease, the positively and negatively skewed stimuli were mirror copies of each other. Cat stimuli had an r.m.s. contrast of 0.7, skews of ±0.4 and a flat power spectrum band-limited to 124 Hz. Salamander stimuli had an r.m.s. contrast of 0.2, skews of ±2 and a flat power spectrum band-limited to 30 Hz.

### Statistics

All data are presented as the mean (SD), unless otherwise stated. Statistical significance was tested with Student's paired or unpaired *t*-test or Welch's *t*-test, as appropriate. The reported *P*-values were adjusted for multiple comparisons with the Benjamini–Hochberg procedure (Benjamini & Hochberg, 1995).

### Results

#### Cone responses vary with skewness

Direct light responses of photoreceptors are not affected by the spatial structure of stimuli and instead depend only on the intensity of light falling on the photoreceptor outer segment (see 'Light stimuli' in the Methods section). Hence, to assess the contribution of the photoreceptor to skew discrimination, we exposed goldfish cones to a series of modified NTSCI from the Van Hateren library (Van Hateren *et al.* 2002; skew stimulus set 1)

and recorded their photocurrent and voltage responses (Fig. 3*A*). Stimuli were equal in terms of mean intensity, r.m.s. and median contrast, and had similar power spectra, and their skewness varied from −2.2 to +2.2. Positively and negatively skewed stimuli were mirror copies of each other; therefore, any asymmetries between corresponding responses would reflect an asymmetry in processing by the cone.

To determine whether cones process negatively and positively skewed light stimuli in a different manner, we plotted the skews of the photocurrent (Fig. 3*B*) and voltage responses (Fig. 3*C*) against the skews of the light stimuli. If there is no difference in processing, the skewness of the response will be equal to the light stimulus skewness, and thus the data points will fall along a straight slope. However, if there is an asymmetry in the processing of positive and negative contrasts it would necessarily lead to a deviation of the data points from the grey line. Figure 3*B* and *C* shows that, for positively skewed stimuli, the photocurrent and voltage responses are skewed to a lesser degree than are the light stimuli, whereas for the negatively skewed stimuli they are almost as equally skewed as the light stimuli (note that the signal sign inversion of the voltage response also sign inverts its skewness).

To quantify this difference in response to positively and negatively skewed stimuli, for each of the three stimuli pairs (i.e. ±0.9, ±1.6 and ±2.2), we determined, for each cone, the absolute difference in skew between the stimulus and response in the positively skewed conditions (|Δ positive skew|) and in the negatively skewed conditions (|Δ negative skew|) and then subtracted the negative skew term from the positive skew term (|Δ positive skew| −

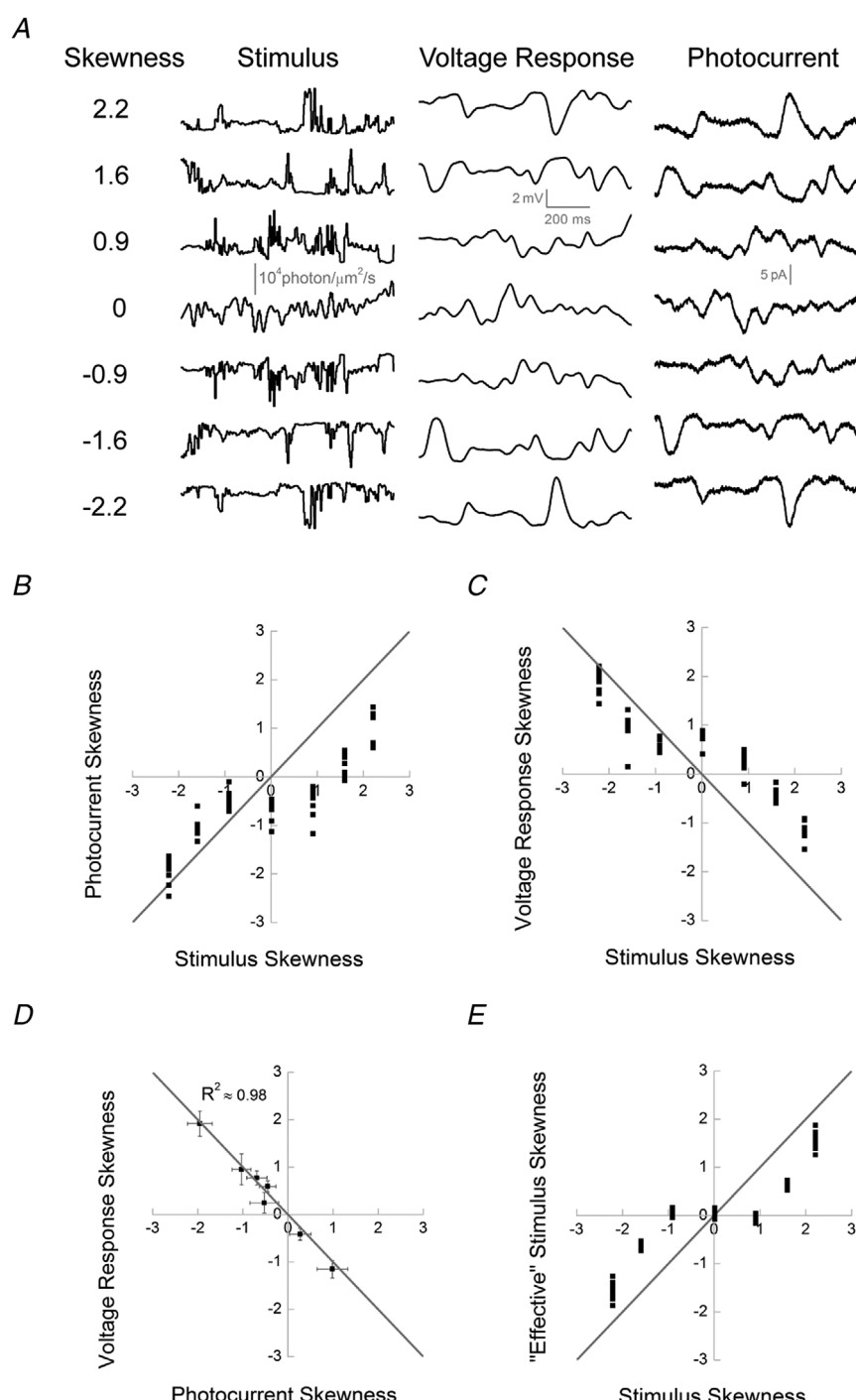

**Figure 3. Cone processing affects stimulus skewness**

*A*, skew stimulus set 1 (left) and examples of voltage (middle) and photocurrent (right) responses from cone photo-receptors. *B*, photocurrent skewness as a function of stimulus skewness for skew stimulus set 1 (*n* = 8). The grey line indicates equal response and stimulus skewness. The graph shows that cone responses to positively skewed stimuli were less skewed than their stimuli and that this effect was weaker for the negatively skewed stimuli [|Δ positive skew| − |Δ negative skew|: ±0.9, 0.97 (0.436), *P* = 0.002; ±1.6, 0.76 (0.419), *P* = 0.004; and ±2.2, 0.97 (0.537), *P* = 0.003; *n* = 8, Student's paired *t*-test]. Also note the zero-skewed stimulus elicited skewed responses. *C*, voltage response skewness as a function of stimulus skewness (*n* = 9). Note that the voltage responses and stimuli skews have different signs owing to the signal sign inversion. The grey line indicates equal response and stimulus skews. As was the case in *B*, cone responses to positively skewed stimuli were less skewed than their stimuli, the effect was weaker for the negatively skewed stimuli, and the zero-skewed stimulus led to a skewed response [|Δ positive skew| − |Δ

negative skew|: ±0.9, 0.84 (0.235), $P = 0.00003$; ±1.6, 0.54 (0.341), $P = 0.004$; and ±2.2, 0.77 (0.280), $P = 0.0001$; $n = 9$, Student's paired $t$-test]. D, mean (±SD) skewness of the voltage response as a function of the mean (±SD) photocurrent skewness. The grey line indicates equal voltage and photocurrent skews. No statistically significant differences were observed between the absolute magnitude of skewness of photocurrent and the voltage response ($P$-values of 0.78, 0.60, 0.14, 0.45, 0.10, 0.23 and 0.30 for the stimulus skew values of −2.2, −1.6, −0.9, 0, 0.9, 1.6 and 2.2, respectively, Welch's test). E, 'effective' skewness as the function of the stimulus skewness. 'Effective' skews were estimated from the convolution product of the light stimuli with the impulse response function of the cone (Fig. 4A and B). The grey line indicates equal 'effective' and response skews. This graph shows that cone temporal filtering leads to a reduction of the stimuli skewness (0, ±0.9, ±1.6 and ±2.2) in both photocurrent [respectively, 0.10 (0.05), $P = 0.003$; ± 0.11 (0.03), $P < 0.0001$; ±0.63 (0.07), $P < 0.0001$; and ±1.54 (± 0.13), $P < 0.0001$, Student's unpaired $t$-test] and voltage response [respectively, 0.03 (0.06), $P = 0.2$; ±0.02 (0.06), $P < 0.0001$; ±0.61 (0.07), $P < 0.0001$; and ±1.57 (0.16), $P < 0.0001$, Student's unpaired $t$-test].

|Δ negative skew|). This showed that cone responses were significantly less skewed, relative to their stimulus, when positively skewed stimuli were used in comparison to when using negatively skewed stimuli for both the photocurrent [|Δ positive skew| − |Δ negative skew|: ±0.9, 0.97 (0.436), $P = 0.002$; ±1.6, 0.76 (0.419), $P = 0.004$; and ±2.2, 0.97 (0.537), $P = 0.003$; $n = 8$, Student's paired $t$-test] and voltage response [|Δ positive skew| − |Δ negative skew|: ±0.9, 0.84 (0.235), $P = 0.00003$; ±1.6, 0.54 (0.341), $P = 0.004$; and ±2.2, 0.77 (0.280), $P = 0.0001$; $n = 9$, Student's paired $t$-test]. Thus, Fig. 3B and C indicates an asymmetry in the processing of negatively and positively skewed stimuli by cone photoreceptors.

## The processing asymmetry originates exclusively within the phototransduction cascade

What are the cellular mechanisms leading to the differences in the processing of negatively and positively skewed stimuli? To tease apart the relative contributions of the phototransduction cascade and the voltage-activated membrane conductances, we plotted the skewness of the voltage responses and photocurrent against each other in Fig. 3D. The grey line depicts the conditions in which photocurrent and voltage response skews are equal in magnitude. All data points fell on this line, and the degree of photocurrent and voltage response skewness did not differ statistically at any stimulus skew level (respectively, for stimulus skew values −2.2, −1.6, −0.9, 0, 0.9, 1.6 and 2.2, Welch's $t$-test $P$-values = 0.78, 0.56, 0.09, 0.40, 0.06, 0.16 and 0.25). This means that the phototransduction cascade is the primary source of the asymmetric processing of the positively and negatively skewed stimuli.

## Temporal filtering affects stimulus skewness

The finite kinetics of a cone act as a linear temporal filter, attenuating the faster changes of light intensities in a stimulus more than the slower changes, which might in itself contribute to the cone response skewness being different from that of the stimulus (Fig. 3B and C). Aspects of the stimulus changing on time scales unavailable or barely accessible to drive cone responses are still included when calculating stimulus skewness. Hence, the skewness calculated for the stimulus and for the portion of the stimulus able to elicit a cone response can differ. A similar issue occurs when calculating contrast (Howlett *et al.* 2017). Importantly, the major part of the linear temporal filtering occurs before the conversion of the chemical signal into changes of the photocurrent, where the literature suggests the asymmetric gain of cone responses originate (van Hateren, 2005). In this subsection, we estimate the stimuli available to drive cone responses after the linear filtering stage, which we term 'effective' stimuli. An 'effective' stimulus can also be thought of as the linear prediction of the response of a cone to the light stimulus.

To determine the 'effective' stimuli skews, we first estimated the temporal filters of the photocurrent and voltage responses of a cone to skew stimulus set 1, following the Wiener approach (Wiener, 1964; Rieke, 2001; Fig. 4). Next, we convolved the estimated filters with the skewed stimuli to obtain the 'effective' stimuli. Then we calculated the skews of the 'effective' stimuli and plotted them against the skews of the original light stimuli in Fig. 3E, where the grey line describes the situation where the 'effective' skew is equal to the original skew. Given that, for each skew value, corresponding positively and negatively skewed stimuli were mirror copies of each other, hence temporal filtering affected the magnitude of their skewness equally. Data points for the positively skewed light stimuli are lower than the grey line and higher for the negatively skewed light stimuli. Hence, in all cases, the skewness of the light stimuli (0, ±0.9, ±1.6 and ±2.2) was greater than the 'effective' skew estimated using either the cone photocurrent [respectively, 0.10 (0.05), $P = 0.003$; ±0.11 (0.03), $P < 0.0001$; ±0.63 (0.07), $P < 0.0001$; and ±1.54 (± 0.13), $P < 0.0001$] or voltage response [respectively, 0.03 (0.06), $P = 0.2$; ±0.02 (0.06), $P < 0.0001$; ±0.61 (0.07), $P < 0.0001$; and ±1.57 (0.16), $P < 0.0001$]. The adjusted $P$-values describe the significance of the difference between 'effective' and 'original' skew stimulus estimated with Student's unpaired $t$-test.

### Asymmetry in the responses to 'effective' stimuli

How do goldfish cones process these 'effective' stimuli? Figure 3*E* shows the 'effective' skew dynamic range available to cone processes after the initial linear temporal filtering stages was reduced by almost 30%, relative to the light stimuli (from ±2.2 to ±1.6). Therefore, we first completed our data set by recording the voltage responses of a cone to stimuli with 'effective' skews of ±2.2 (Figs 4*B* and 5*A*).

Next, we plotted the skews of the responses against the 'effective' stimulus skews (Fig. 5*B*) and found that goldfish cones decrease the magnitude of skewness when the stimuli are skewed positively and increase the magnitude of skewness when the stimuli are skewed negatively. Asymmetry in the processing of negatively and positively skewed stimuli is also reflected in the amplitudes of the corresponding responses; standard deviations of the responses to negatively skewed stimuli were ≤50% larger than the standard deviations of the responses to positively skewed stimuli (Fig. 5*C*).

### Asymmetry in the output of a cone

To be relayed to the downstream neurons, asymmetries in the responses of the cone to positively and negatively

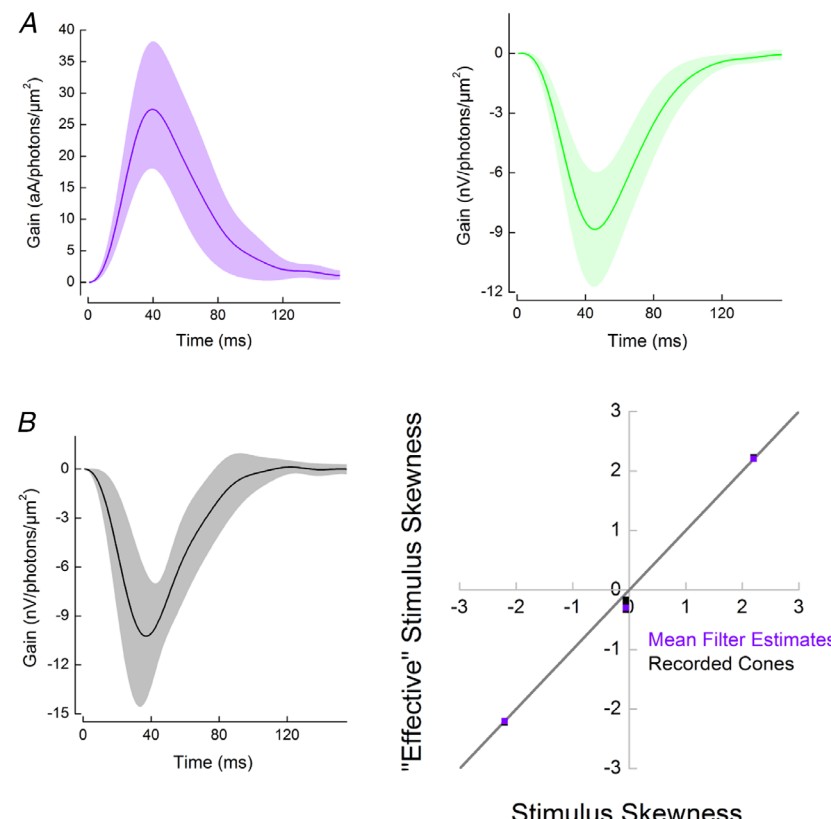

**Figure 4. Temporal filtering by cones does not change the skewness of stimuli from skew stimulus set 2**
*A*, the mean cone impulse response functions (±SD) estimated from the photocurrent responses (violet, *n* = 8) and from the voltage responses (green, *n* = 9) to skew stimulus set 1. Individual impulse response functions calculated for each cone were used to estimate the levels of 'effective' skew delivered by skew stimulus set 1 (Fig. 3*E*). The mean impulse response function of the photocurrent was used during the selection process for skew stimulus set 2. *B*, left panel, the mean cone impulse response function (±SD) estimated from the voltage responses to for skew stimulus set 2 (*n* = 7). Right panel, skewness of the 'effective' stimuli as a function of the skewness of the light stimuli for skew stimulus set 2. Violet squares correspond to the 'effective' skewness obtained by convolving the light stimuli with the mean photocurrent impulse response function estimated from responses to skew stimulus set 1 shown in *A*. Black squares depict the 'effective' skewness of each cone response recorded under skew stimulus set 2. This was estimated by convolving each skew stimulus set 2 stimulus with the cone impulse response functions used to calculate *B*, left panel. Note that to simplify the visualization of the right panel of *B*, the cone response 'effective' skews (black) were multiplied by minus one. The grey line describes the condition where temporal filtering does not affect stimulus skewness. Given that all squares are aligned with the grey line, the 'effective' skewness is approximately equal to the original light stimulus skewness [for zero skew, 'effective' skews were 0.26 (0.06); for the ±2.2 skews, 'effective' skews were ±2.2 (0.01)]. [Colour figure can be viewed at wileyonlinelibrary.com]

skewed stimuli (Fig. 5*B* and *C*) should be reflected in the synaptic release. In photoreceptors, glutamate release is directly proportional to $I_{Ca}$ (Schmitz & Witkovsky, 1997; Thoreson *et al.* 2004). Consequently, one can estimate changes in cone glutamate release by recording its $I_{Ca}$.

We measured skew-dependent modulation of $I_{Ca}$ by using recorded voltage responses to light stimuli with

'effective' skews of ±2.2 and −0.3 as the command voltages (Fig. 6*A*) at three different potentials (−30, −40 and −50 mV) along the $I_{Ca}$ activation curve. To isolate $I_{Ca}$ responses, we blocked all other active conductance and subtracted the leak current. We then plotted the skews of the $I_{Ca}$ signals against the skews of the voltage responses (Fig. 6*B*). Note that as depolarization produces an inward

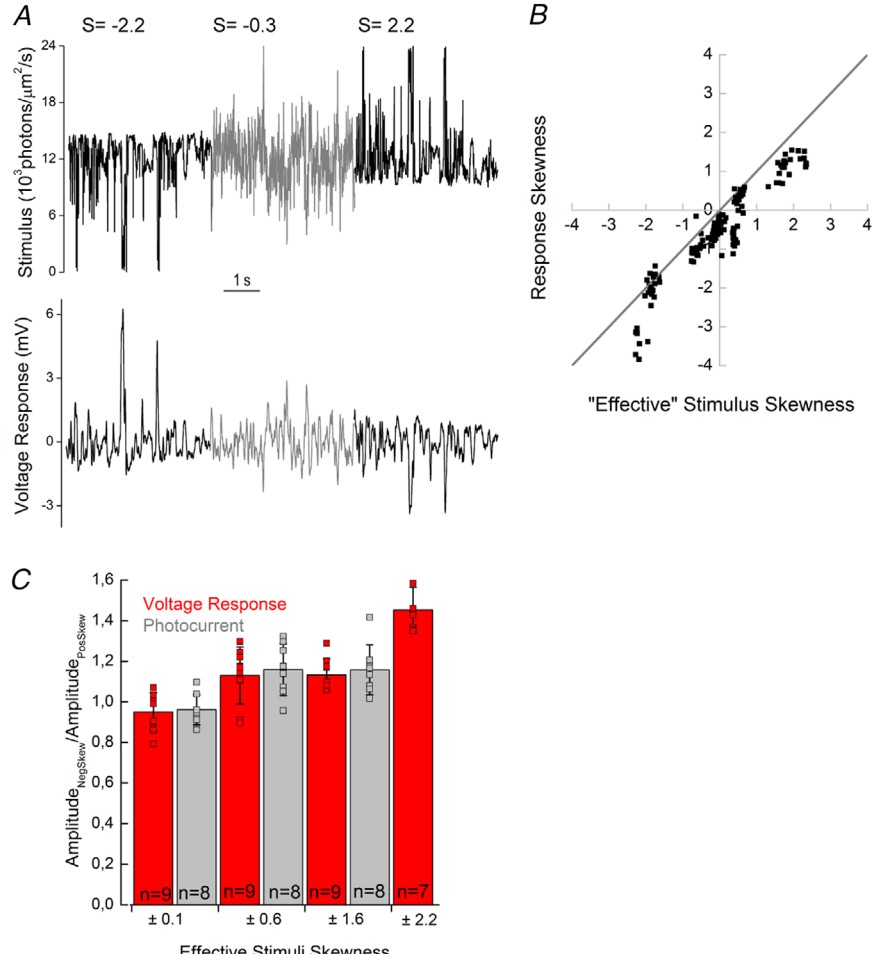

**Figure 5. Asymmetries in cone responses to positively and negatively skewed stimuli**
*A*, top trace, skew stimulus set 2; bottom trace, a recorded voltage response. *B*, skews of voltage and photocurrent responses plotted against the 'effective' stimuli skews. This graph illustrates that cone responses are less skewed than the stimulus when the stimulus delivers higher levels of positive 'effective' skews and more skewed when high levels of negative 'effective' skews are delivered. The grey line depicts equal cone response and 'effective' stimulus skews. For illustrative convenience, the voltage response skews were multiplied by minus one. *C*, differences in cone response amplitudes (grey, photocurrent; red, voltage response) to negatively and positively skewed stimuli for each 'effective' skew stimulus pair. Bar height represents the mean value, black lines the SD, and squares the individual cells. When the 'effective' skew magnitude was low (±0.1), the photocurrent or voltage response amplitudes were unaffected by the skew direction. However, at higher 'effective' skew magnitudes, the cone response amplitudes to negatively skewed stimuli were larger than for positively skewed stimuli [photocurrent difference: ±0.1: −3.9 (7.39)%, $P = 0.23$; ±0.6: 15.8 (12.80)%, $P = 0.02$; and ±1.6: 15.7 (12.26)%, $P = 0.02$; voltage response difference: ±0.1: −5 (9.5)%, $P = 0.22$; ±0.6: 13.0 (14)%, $P = 0.051$; ±1.6: 13.3 (7.92)%, $P = 0.003$; and ±2.2, 45.2 (11.17)%, $P = 0.0001$, Student's paired *t*-test]. Changes in response amplitude were assessed as the ratio of standard deviations of the response of a cell to corresponding negatively and positivity skewed stimuli. Note that, for convenience, the 'effective' skew values shown are the average estimates of the 'effective' skews for the photocurrent and voltage responses to skew stimulus set 1, rounded to the first decimal point. [Colour figure can be viewed at wileyonlinelibrary.com]

$I_{Ca}$, the skews of the voltage response and $I_{Ca}$ response have opposite signs. Regardless of the clamping potential, all the data points in Fig. 6*B* fall approximately on the grey line, indicating that the skewness of the signal from the cone is largely unaffected by the transformation from the membrane potential to the $I_{Ca}$.

Next, we tested whether the $I_{Ca}$ amplitudes elicited by the +2.2 and −2.2 skew conditions differed more or less, relative to how much the used voltage commands differed in amplitude (45.8%). At holding potentials of −30 mV [49.5 (3.02)%, $n = 6$, $P = 0.07$, Student's unpaired *t*-test] and −50 mV [50.7 (4.9)%, $n = 6$, $P = 0.066$, Student's unpaired *t*-test], there were no statistically significant changes in the amplitude ratio. However, when held at −40 mV, the difference in $I_{Ca}$ amplitudes [54.4 (2.58)%, $n = 6$, $P = 0.005$, Student's unpaired *t*-test] become more pronounced than in the voltage stimulus. Thus, for negatively and positively skewed stimuli, the asymmetries present at the earlier processing stages by the cone are preserved and even somewhat enhanced in the output of the cone.

## Asymmetric gain of the cone photoreceptors

What type of non-linear gain leads to the skew-dependent changes in the amplitude of the cone responses? To determine how the amplitude of the voltage response depends on the Weber contrast step, we first converted the 'effective' stimulus intensities into Weber contrast steps (Fig. 7). Next, we plotted baseline subtracted mean voltage responses (Fig. 7) as a function of the 'effective' Weber contrast steps (Fig. 8*A*). Figure 8*A* shows that cone responses are larger for Weber contrast steps below −0.4 than they are for Weber contrast steps above 0.4. Hence, the response gain of cones is greater for high negative contrasts than for high positive contrasts.

What type of input–output relationship supports the asymmetric gain of cones? We fitted the relationship between voltage responses and 'effective' Weber contrast steps with Eq. (3) (Fig. 8*A*, orange line) and found that the cone voltage response is proportional to the Weber contrast step with an exponent of −0.1251 ($R^2 \approx 0.95$, 95% confidence interval ± 0.0959). This dependence is

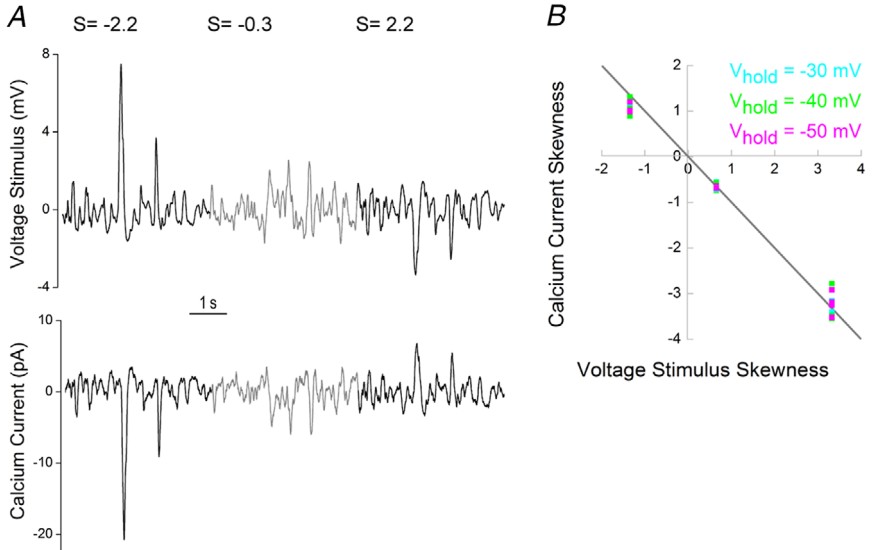

**Figure 6. Asymmetric processing of skewed stimuli at the cone $I_{Ca}$ level**
*A*, top trace depicts the mean cone voltage response to skew stimulus set 2; bottom trace shows the resulting $I_{Ca}$. *B*, skewness of $I_{Ca}$ as a function of voltage stimulus skewness. The measurements of $I_{Ca}$ were performed at three different holding potentials ($V_{hold}$) along its activation curve: −30 mV (cyan), −40 mV (green) and −50 mV (magenta). Note that because a decrease in voltage causes an increase in $I_{Ca}$, skews for the stimulus and response have opposite signs. The grey line indicates equal $I_{Ca}$ and voltage stimulus skews. The light stimulus with an 'effective' skew of −2.2 led to a mean voltage response with a skewness of 3.32, which, when used as a voltage command stimulus, yielded $I_{Ca}$ with skews of: −3.35 (0.144), $P = 0.75$ at $V_{hold} = −30$ mV; −3.37 (0.294), $P = 0.69$ at $V_{hold} = −40$ mV; and −3.27 (0.229), $P = 0.72$ at $V_{hold} = −50$ mV. The light stimulus with an 'effective' skew of −0.3 led to a mean voltage response with a skewness of 0.65, which yielded $I_{Ca}$ with skews of: −0.67 (0.060), $P = 0.7$ at $V_{hold} = −30$ mV; −0.69 (0.060), $P = 0.25$ at $V_{hold} = −40$ mV; and −0.69 (0.030), $P = 0.059$ at $V_{hold} = −50$ mV. The light stimulus with an 'effective' skew of 2.2 led to a mean voltage response with a skewness of −1.35, which yielded $I_{Ca}$ with skews of: 1.2 (0.05), $P = 0.005$ at $V_{hold} = −30$ mV; 1.12 (0.162), $P = 0.07$ at $V_{hold} = −40$ mV; and 1.12 (0.092), $P = 0.007$ at $V_{hold} = −50$ mV; $n = 6$ in all cases. Adjusted *P*-values reflect the significance of the difference between the absolute magnitude of skews of the voltage and $I_{Ca}$ responses tested with Student's unpaired *t*-test. Differences between the absolute magnitude of skews for the voltage command stimulus and the $I_{Ca}$ responses were tested with Student's unpaired *t*-test. [Colour figure can be viewed at wileyonlinelibrary.com]

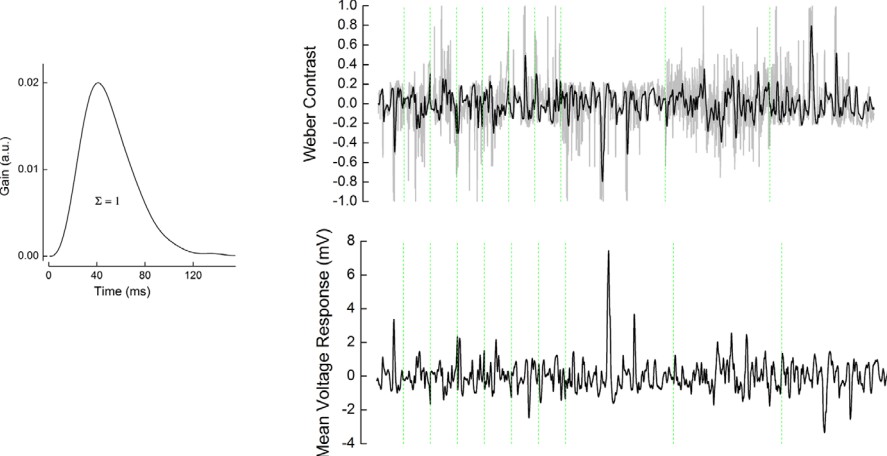

**Figure 7. 'Effective' Weber contrast steps**
Left panel, mean impulse response function used to obtain 'effective' Weber contrast. The mean impulse response was estimated by averaging the individual voltage response impulse response functions of all cells for all stimuli ($n$ = 16; Fig. 4*A*, right panel, and Fig. 8*B*). This mean was scaled such that the integral under its curve was one. Right panel, top, grey line indicates the original light stimuli of skew stimulus sets 1 and 2 converted to Weber contrast steps. Black line indicates the 'effective' Weber contrast steps obtained by the convolution of the original Weber contrast steps with the mean impulse response function shown on the left. Right panel, bottom, the averaged cone voltage response to each set of stimuli. For both top and bottom panels, the green dotted lines separate the different stimuli stretches during which the effective skews were (from left to right): −1.6, −0.1, 0.1, −0.6, 0.07, 1.6, 0.6, −2.2, −0.3 and 2.2. [Colour figure can be viewed at wileyonlinelibrary.com]

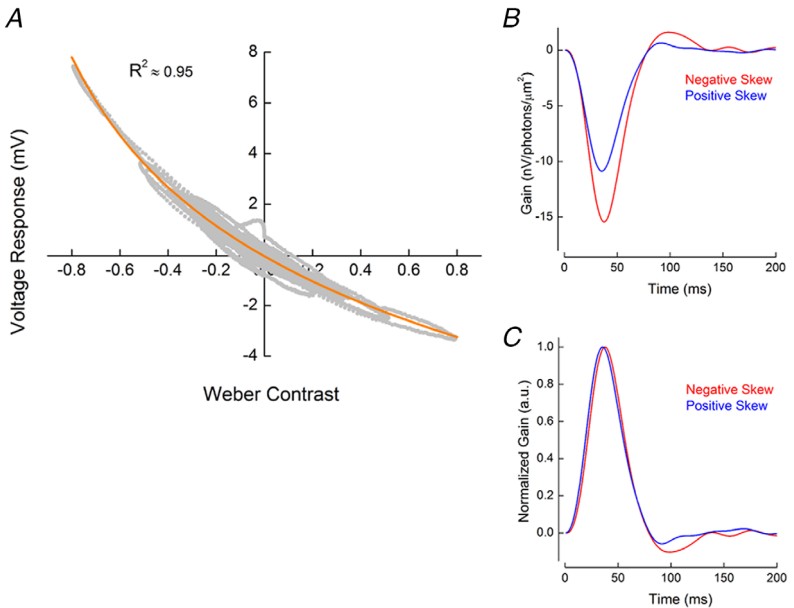

**Figure 8. The cone signal transfer properties**
*A*, asymmetric gain of cone photoreceptors. To estimate cone photoreceptor gain, we plotted the voltage response of cones as a function of 'effective' Weber contrast (Fig. 7). This relationship was well described (orange line, adjusted $R^2$ = 0.95) by the Weber contrast power function given in Eq. (3). *B*, stimulus skewness changes the shape of the impulse response function of the cone. The graph depicts two representative examples of the impulse response function of a cone in conditions with −2.2 (red) and +2.2 (blue) 'effective' stimulus skewness. *C*, the cone impulse response functions shown in *B*, normalized by the amplitude of their initial lobe. On average, impulse response functions peaked 3.6 ms, or 9 (2.6)%, later for the negatively skewed stimulus than for the positively skewed conditions ($P$ = 0.0002, $n$ = 7), whereas the impulse response function full width at half-maximum [FWHM; $\Delta$ = 1(3.6)%, $P$ = 0.49, $n$ = 7] and the cone integration time [$\Delta$ = 1.1(4.26)%, $P$ = 0.51, $n$ = 7] were unchanged. Statistical significance was assessed with Student's paired *t*-test. [Colour figure can be viewed at wileyonlinelibrary.com]

close to the one determined in the theoretical study by Van Hateren & Snippe (2006), who suggested that voltage responses of the vertebrate cone are proportional to the Weber contrast with an exponent of −0.12.

### Stimulus skewness affects the shape of the cone impulse response function

The processing time of cones is inversely proportional to light intensity such that responses to steps of strong positive contrast peak earlier than responses to strong negative contrasts (Nikonov *et al.* 2000; Lee *et al.* 2003; van Hateren, 2005; Van Hateren & Snippe, 2006; Angueyra *et al.* 2021). Therefore, one might expect that such a dependence would lead to a difference in the time courses of the responses to positively and negatively skewed stimuli.

To test whether stimulus skewness has an effect on the cone kinetics, we compared impulse response functions derived from the voltage responses to the −2.2 and the +2.2 'effective' skew stimuli (Fig. 8*B*). Impulse response functions estimated in such a way might differ from the true stimulus–response function of the system because of the biases associated with the skewness of the stimuli. However, given that the positively and negatively skewed stimuli in skew stimulus set 2 differed only in terms of their skewness, the comparison of the corresponding impulse response function still enables the assessment of any potential effect of the stimulus skewness on the signal-processing properties of the cone.

To visualize the differences in kinetics better, we normalized these impulse response functions by the amplitude of their initial lobe (Fig. 8*C*). Interestingly, on average the cone impulse response functions peaked 3.6 ms [or 9 (2.6)%] later for the negatively skewed stimulus than for the positively skewed stimulus (Fig. 8*C*; $P = 0.0002$, $n = 7$), whereas there were no statistically significant differences in either its FWHM [$\Delta = 1$ (3.6)%, $P = 0.41$; $n = 7$] or in the integration time of the cone [$\Delta = 1.1$ (4.26)%, $P = 0.51$; $n = 7$]. For the FWHM and integration time to remain unchanged while the time to peak shifts suggests that during negatively skewed stimuli the reduced rise time of the initial flank of the impulse response function is largely offset by the peak falling back to baseline at a faster rate.

### Discussion

We studied responses of cone photoreceptors to differently skewed stimuli and found that cone response amplitudes to negatively skewed stimuli are ≤50% greater than to positively skewed stimuli (Figs 2, 5*A* and *C*, 6*A* and 7). This difference in amplitude originates from the asymmetric weighting of positive and negative contrasts by the phototransduction cascade. Its gain is inversely

proportional to Weber contrast steps raised to the power of −0.125 (Van Hateren & Snippe, 2006; Fig. 8*A*) and might serve as the basis for the blackshot mechanism proposed by Chubb *et al.* (1994, 2004). Additionally, we observed that stimulus skewness changes the shape of the impulse response function of the cone. For the normalized impulse response function, the rising flank was faster and the falling flank slower for positively compared with negatively skewed stimuli (Fig. 8*C*).

### The blackshot mechanism

Psychophysical studies reported that humans can discriminate visual stimuli based on skewness (Chubb *et al.* 1994, 2004; Graham *et al.* 2016). Chubb *et al.* (1994, 2004) described the sensitivity to skewness with the so-called blackshot mechanism, which has a disproportionally strong response to high negative contrasts. Our results indicate that, for goldfish retina, discrimination between skewed stimuli, and the processing asymmetry that enables it, start as early as the phototransduction cascade (Figs 5*C* and 8*A*). Given the highly conserved nature of photoreceptor signal processing across vertebrate retina (see review by Fain *et al.* 2010), our results potentially suggest that the ability of human subjects to discriminate visual stimuli based on skewness might also stem from the asymmetric gain of the phototransduction cascade, which enhances strong negative contrast more than strong positive contrast. Additional support for this idea might also come from the qualitative similarities between psychophysical studies (Chubb *et al.* 1994, 2004) and our data, showing that the difference in cone response amplitudes to positive and negative contrasts becomes more prominent with larger contrast steps (Fig. 8*A*). For the ±1.6 'effective' skew stimuli pair, where the maximal 'effective' Weber contrast step was 0.5, the difference in the response was ∼15%, whereas for the ±2.2 'effective' skew pair, where the maximal 'effective' Weber contrast step was 0.8, the difference was almost 50% (Fig. 5*C*).

Asymmetries in responses to positive and negative contrasts are reported throughout the entire visual system in various species (Laughlin, 1981; Van Hateren, 1997; Lee *et al.* 2003; Zaghloul *et al.* 2003; Jin *et al.* 2008; Yeh *et al.* 2009; Endeman & Kamermans, 2010; Baden *et al.* 2013; Kremkow *et al.* 2014; Cooper & Norcia, 2015), including the human visual cortex (Zemon *et al.* 1988; Kremkow *et al.* 2014). Although it was shown that differences in response amplitudes to positive and negative contrasts are additionally amplified by the visual cortex (Kremkow *et al.* 2014), our data clearly indicate that the primary origin of this asymmetry is within the phototransduction cascade of the cone.

The asymmetric gain function of phototransduction enables cones efficiently to encode the entire range of

contrasts present in natural scenes. Photoreceptors encode changes in their input with a graded output. Information theory states that such a system encodes a signal efficiently only when the statistical distribution of its output is Gaussian, which implies a symmetric engagement of the dynamic range of the system (Shannon, 1948; Van Hateren, 1997). In contrast, from a given mean, the light intensity cannot decrease by >100%, but it can easily increase by many orders of magnitude. This means that the dynamic range of positive contrasts is wider than that of negative contrasts. Thus, although some visual scenes can be skewed negatively (Tkačik *et al.* 2014), the total distribution of contrasts at any given intensity is skewed positively, with negative contrasts being smaller in amplitude but more frequent than positive contrasts (Laughlin, 1983; Ruderman, 1994; Van Hateren, 1997; Ruderman *et al.* 1998; Cooper & Norcia, 2015). Consequently, to encode signals efficiently and to provide symmetric outputs, cones compensate for this asymmetry in their input by weighting high positive contrasts with lower gain (Fig. 8*A*), such that, when stimulated with the entire range of contrasts in natural scenes, cones provide a Gaussian output (Laughlin, 1983; Van Hateren, 1997; Endeman & Kamermans, 2010). We therefore suggest that the blackshot mechanism is simply a consequence of the more fundamental necessity to encode the range of contrasts present in natural scenes efficiently.

It is also important to note that although, when it comes to dealing with wide dynamic ranges of light intensities, the non-linear gain of photoreceptors is well acknowledged, its influence on perception is often ignored such that responses of the downstream neurons are often modelled with an implicit assumption of photoreceptor linearity (Chander & Chichilnisky, 2001; Kim & Rieke, 2001; Rieke, 2001; Carandini *et al.* 2005; Mante *et al.* 2005; Manookin & Demb, 2006; Ozuysal & Baccus, 2012; Pitkow & Meister, 2012; Karamanlis & Gollisch, 2021; Schreyer & Gollisch, 2021). However, there are a number of occasions when this non-linearity is important to account for perceptual features (Kremkow *et al.* 2014; Angueyra *et al.* 2021). For instance, higher spatial resolution for darker than for brighter patches is a consequence of the cone photoreceptor non-linear gain (Kremkow *et al.* 2014). Our data present another example of the far-reaching consequences of the early visual non-linearity and highlight the importance of accounting for the photoreceptor non-linearity when studying the visual system.

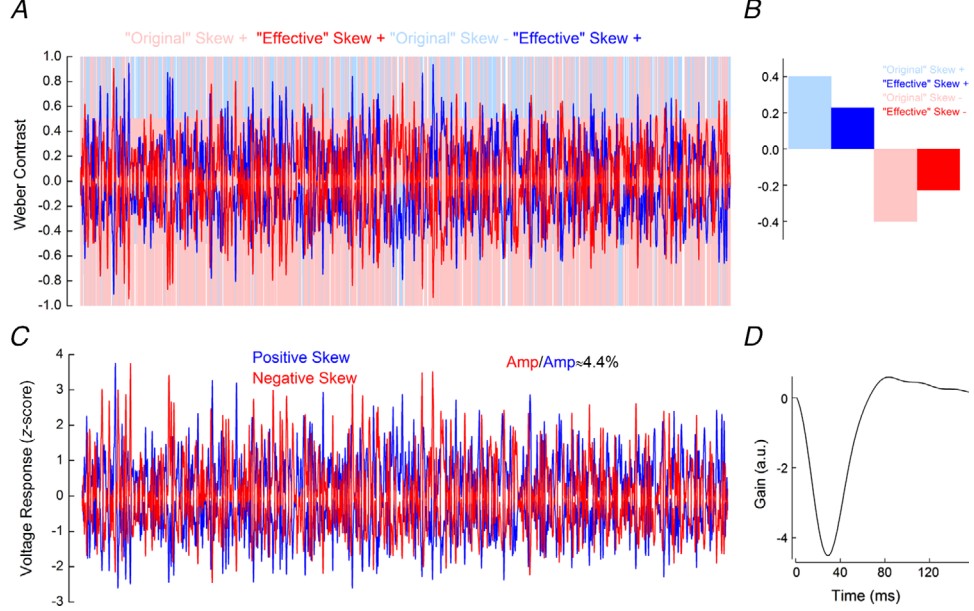

**Figure 9. Simulated responses of cat cone photoreceptors**
*A*, light blue and light red lines depict the positively and negatively skewed stimuli used by Bonin *et al.* (2006) and here for the simulation. Blue and red lines depict the 'effective' positively and negatively skewed stimuli, obtained by the convolution of the 'original' stimuli with the impulse response function shown in *D*. *B*, comparison of the skews of the 'original' and 'effective' positively (blue) and negatively (red) skewed stimuli. Owing to temporal filtering, the 'effective' stimuli are almost symmetric around the mean, their skew range having decreased from ±0.4 to approximately ±0.2. *C*, simulated cone voltage responses to the positively (blue) and negatively (red) skewed stimuli. The response amplitude to the negatively skewed stimulus was 4% higher than the response amplitude to the positively skewed stimulus. The simulated voltage response amplitudes were quantified by their standard deviations, as for Fig. 5*C*. Parameters for the simulation are listed in Table 1. *D*, impulse response function derived from the simulated cat cone voltage responses. This impulse response function was used to estimate the 'effective' stimuli shown in *A*. [Colour figure can be viewed at wileyonlinelibrary.com]

## The role of the non-linear membrane conductances in the asymmetric processing of strong positive and negative contrasts

Figure 3D indicates that the light–dark asymmetry originates within the phototransduction cascade. This might be rather surprising considering that there are several non-linear membrane conductances in cone photoreceptors. For instance, Barrow & Wu (2009) showed that the hyperpolarization-activated current, $I_h$, can compress the photoreceptor response to light flashes. We propose that this apparent contradiction originates from the differences in stimulus conditions. Barrow and Wu delivered their bright light flashes from darkness, and such conditions lead to extremely high levels of positive contrast, whereas in the present study the positive contrast was never >0.8. It might be that in such conditions the degree of $I_h$ activation is too weak to produce any measurable effect on the response skewness. This is also consistent with the fact that S-cones lack a prominent $I_h$ (Howlett *et al.* 2017) but do not appear as outliers in Figs 3C and 5B. Indeed, given that $I_h$ compresses responses to strong positive contrasts, one should expect that for M- and L-cones its activation will decrease skewness of the voltage responses to positively skewed light stimuli to a larger extent than for S-cones, and this was not the case. However, a meaningful statistical analysis to support this claim is not possible because we recorded from only a small number of S-cones (two for the each of the stimulus sets).

Although we did not observe any contribution of voltage-activated membrane conductances to the asymmetric voltage response of the cone to negatively and positively skewed stimuli, we did find this response asymmetry to be affected by voltage-activated calcium channels at the $I_{Ca}$ level. Indeed, at a holding potential of −40 mV, the $I_{Ca}$ response amplitude asymmetry was 8.6 (2.6)% greater than it was for the voltage response. Thus, the asymmetric cone processing of strong positive and negative contrasts that originates within the phototransduction cascade also undergoes some minor amplification during the transformation from a voltage response to an $I_{Ca}$ signal.

## Shape of the impulse response function

We found the impulse response function of the cone peaks ~3.6 ms later for negatively skewed stimuli, whereas the integration time of the cone is unaffected by stimulus

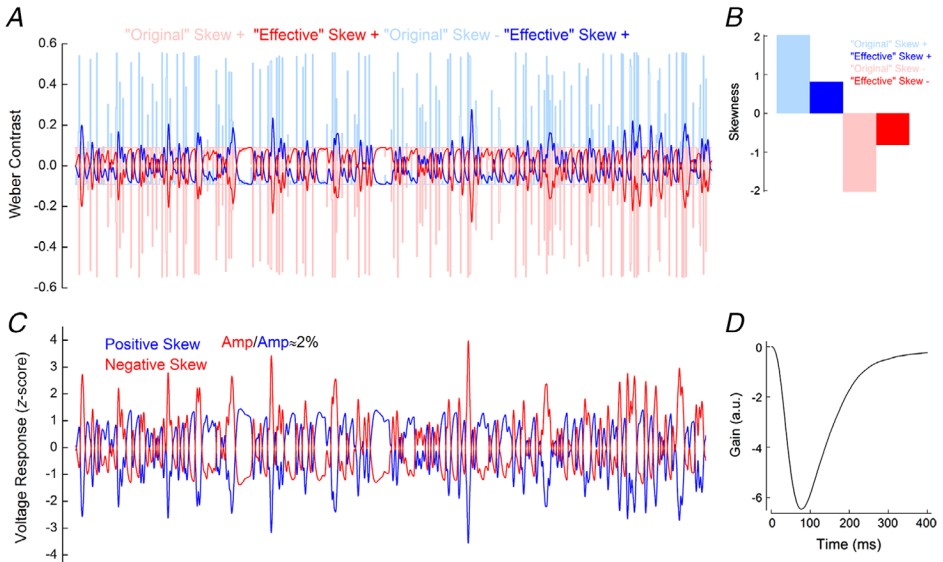

**Figure 10. Simulated responses of salamander cone photoreceptors**
*A*, light blue and light red lines depict the positively and negatively skewed stimuli used in the simulation. These stimuli had the same properties as those used by Tkačik *et al.* (2014). Blue and red lines depict the 'effective' positively and negatively skewed stimuli, obtained by the convolution of the 'original' stimuli with the impulse response function shown in *D*. The temporal filtering leads to small Weber contrast ranges of the 'effective' stimuli. *B*, comparison of the skews of the 'original' and 'effective' positively (blue) and negatively (red) skewed stimuli. The temporal filtering of the salamander cone reduced the stimulus skewness from ±2 to ±0.8. *C*, simulated cone voltage responses to the positively (blue) and negatively (red) skewed stimuli. The response amplitude to the negatively skewed stimulus was 2% higher than the response amplitude to the positively skewed stimulus. The simulated voltage response amplitudes were quantified by their standard deviations, as for Fig. 5C. Parameters for the simulation are listed in Table 1. *D*, impulse response function derived from the simulated salamander cone voltage responses. This impulse response function was used to estimate the 'effective' stimuli shown in *A*. [Colour figure can be viewed at wileyonlinelibrary.com]

skewness (Fig. 8*B* and *C*). The rising and falling flanks of the impulse response of the cone are governed by different biophysical mechanisms. The former is heavily influenced by the hydrolysis of cGMP by PDE. The time constant of this process is inversely proportional to the light intensity. Hence, it decreases upon positive contrast steps and increases upon negative contrast steps (Nikonov *et al.* 2000; van Hateren, 2005; Endeman & Kamermans, 2010). The terminating flank is largely regulated by the GC-mediated production of cGMP, which is modulated by $Ca^{2+}$ influx through the cyclic-nucleotide gated (CNG) channels. The time constant of this process is not light dependent, but its gain is inversely proportional to the fourth power of light intensity (Burns *et al.* 2002; van Hateren, 2005; van Hateren & Snippe, 2007). The interplay between these two underlying processes is thought to account for the shape of the impulse response function of the cone, for the asymmetric rising and falling response phases to sinusoidal stimuli, and for light adaptation to decreases in light intensity being slower than for increases in light intensity (Baylor & Hodgkin, 1973; Lankheet *et al.* 1991; Nikonov *et al.* 2000; Lee *et al.* 2003; van Hateren, 2005; Endeman & Kamermans, 2010; Angueyra *et al.* 2021).

For our stimuli, the time constant of hydrolysis of cGMP by PDE will have been shorter during positively skewed stimuli because all large changes in light intensity were associated with positive contrasts. This, in turn, manifested as a faster rate of change in the initial flank of the impulse response functions, hence an earlier time to peak. For negatively skewed stimuli, given that all large changes in light intensity were associated with negative contrast, GC-mediated production of cGMP was pronounced. The highly non-linear light-dependent gain function of this process, and the ensuing $Ca^{2+}$ influx when cones depolarized, increased the rate at which the cone CNG channels reopened. This resulted in an increased decay rate for the terminating flank of the impulse response function. Hence, the faster onset rate of the initial flank during the positively skewed stimulus is largely offset by the faster decay rate of the terminating flank during the negatively skewed stimulus. Consequently, the cone impulse response function integration time to positive and negative skewed stimuli does not differ, whereas its time to peak does.

### Relationship to previous studies using skewed stimuli

Why did previous studies find that stimulus skewness had little to no effect on retinal ganglion cells (Tkačik *et al.* 2014) and LGN neurons (Bonin *et al.* 2006)? We suggest a methodological factor. In both studies, a large proportion of the stimulus power spectrum was outside the temporal frequency bandwidth of the cone. Indeed,

Bonin *et al.* (2006) used white-noise stimuli band-limited to 124 Hz to study responses of cat LGN neurons, although the cat visual system barely responds to frequencies >32 Hz (Shapley & Victor, 1978; Mante *et al.* 2005). Likewise, Tkačik *et al.* (2014) studied salamander retinal ganglion cell responses with white noise band-limited to 30 Hz, whereas the salamander retina hardly reacts to frequencies >10 Hz (Kim & Rieke, 2001). Thus, in both these studies a large part of their stimuli were 'filtered out', and the remaining 'effective' stimuli were able to elicit only marginal skew-dependent effects.

To illustrate this point, we estimated the 'effective' stimuli delivered by Bonin *et al.* (2006) and Tkačik *et al.* (2014) using Van Hateren's cone photoreceptor model (Van Hateren & Snippe, 2007). Simulations of the cat cone indicate that although a wide range of 'effective' Weber contrasts were present (Fig. 9*A*), the 'effective' skewness was approximately half that of the original stimuli used, reducing from a range of ±0.4 to approximately ±0.2 (Fig. 9*B*). Hence, both stimuli delivered largely similar distributions of 'effective' Weber contrasts. As a result, the simulated voltage responses to the positively and negatively skewed stimuli differed in amplitude by only ∼4% (Fig. 9*C* and *D*), which is within the range of estimates of the standard error for the amplitude differences we find here (Fig. 5*C*).

Simulations of the salamander cone reveal a different situation that, nonetheless, leads to the same outcome. The 'effective' skewness range remained relatively large despite being less than half that of the original stimuli used (±0.8 *vs.* ±2; Fig. 10*B*), but the range of 'effective' contrasts reduced to merely ±0.2 Weber units (Fig. 10*A*). Over this limited range of contrasts, the cone photoreceptor gain is mostly symmetric (Fig. 8*A*) and, as such, the voltage response amplitude to the negatively skewed stimulus was only 2% higher than for the positively skewed stimulus (Fig. 10*C* and *D*). Hence, although the stimuli had substantially different distributions of 'effective' Weber contrasts, the range of contrast values they delivered was too narrow to generate a notable effect.

To conclude, our results show that to study visual processes under varying skewness conditions, the stimuli must be able to deliver sufficient levels of 'effective' skewness over sufficient ranges of 'effective' Weber contrasts.

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

## Additional Information

### Data availability statement

The data that support the findings of this study are available from the corresponding author upon reasonable request.

### Competing interests

The authors declare no competing interests.

### Author contributions

M.Y., M.H. and M.K. designed the study. M.Y. performed experiments and analysis. M.Y., M.H. and M.K. wrote the paper. All authors approved the final version of the manuscript and agree to be accountable for all aspects of the work in ensuring that questions related to the accuracy or integrity of any part of the work are appropriately investigated and resolved. All persons designated as authors qualify for authorship, and all those who qualify for authorship are listed.

### Funding

This work was supported by a grant from the Netherlands Ortagization for Sceintific Research via ZonMW (91215062, to M.K.), a grant from Horizon 2020: 'Switchboard' (M.K.), a grant from ODAS (M.K.) and a grant of the Foundation of Friends of the Netherlands Institute for Neuroscience (NIN) (M.K.).

### Keywords

light/dark processing asymmetry, non-linear gain, photo-receptors, retina, skewness, blackshot mechanism

### Supporting information

Additional supporting information can be found online in the Supporting Information section at the end of the HTML view of the article. Supporting information files available:

**Statistical Summary Document**
**Peer Review History**

