## [Peer Review History · The Journal of Physiology]

Enhancing the dark side: Asymmetric gain of cone photoreceptors underpins their discrimination of visual scenes based on their skewness

Matthew Yedutenko, Marcus H.C. Howlett, and Maarten Kamermans

DOI: 10.1113/JP282152

Corresponding author(s): Maarten Kamermans (m.kamermans@nin.knaw.nl)

The following individual(s) involved in review of this submission have agreed to reveal their identity: Wallace B Thoreson (Referee #1); Christopher L Passaglia (Referee #2)

Review Timeline:

Submission Date:	12-Jul-2021
Editorial Decision:	10-Aug-2021
Revision Received:	21-Sep-2021
Editorial Decision:	13-Oct-2021
Revision Received:	26-Oct-2021
Accepted:	11-Nov-2021

Senior Editor: Ian Forsythe

Reviewing Editor: William Taylor

Transaction Report:

Dear Professor Kamermans,

Re: JP-RP-2021-282152 "Enhancing the darkside: Asymmetric gain of cone photoreceptors underpins discrimination of visual scenes based on their skewness" by Matthew Yedutenko, Marcus H.C. Howlett, and Maarten Kamermans

Thank you for submitting your manuscript to The Journal of Physiology. It has been assessed by a Reviewing Editor and by 2 expert Referees and I am pleased to tell you that it is considered to be acceptable for publication following satisfactory revision.

The reports are copied at the end of this email. Please address all of the points and incorporate all requested revisions, or explain in your Response to Referees why a change has not been made.

NEW POLICY: In order to improve the transparency of its peer review process The Journal of Physiology publishes online as supporting information the peer review history of all articles accepted for publication. Readers will have access to decision letters, including all Editors' comments and referee reports, for each version of the manuscript and any author responses to peer review comments. Referees can decide whether or not they wish to be named on the peer review history document.

Authors are asked to use The Journal's premium BioRender (<https://biorender.com/>) account to create/redrawn their Abstract Figures. Information on how to access The Journal's premium BioRender account is here: <https://physoc.onlinelibrary.wiley.com/journal/14697793/biorender-access> and authors are expected to use this service. This will enable Authors to download high-resolution versions of their figures.

I hope you will find the comments helpful and have no difficulty returning your revisions within 4 weeks.

Your revised manuscript should be submitted online using the links in Author Tasks Link Not Available.

Any image files uploaded with the previous version are retained on the system. Please ensure you replace or remove all files that have been revised.

REVISION CHECKLIST:

- Article file, including any tables and figure legends, must be in an editable format (eg Word)
- Abstract figure file (see above)
- Statistical Summary Document
- Upload each figure as a separate high quality file
- Upload a full Response to Referees, including a response to any Senior and Reviewing Editor Comments;
- Upload a copy of the manuscript with the changes highlighted.

- A potential 'Cover Art' file for consideration as the Issue's cover image;
- Appropriate Supporting Information (Video, audio or data set https://jp.msubmit.net/cgi-bin/main.plex?form_type=display_requirements#supp).

To create your 'Response to Referees' copy all the reports, including any comments from the Senior and Reviewing Editors, into a Word, or similar, file and respond to each point in colour or CAPITALS and upload this when you submit your revision.

I look forward to receiving your revised submission.

If you have any queries please reply to this email and staff will be happy to assist.

Yours sincerely,

Ian D. Forsythe

Deputy Editor-in-Chief
The Journal of Physiology
<https://jp.msubmit.net>
<http://jp.physoc.org>
The Physiological Society
Hodgkin Huxley House
30 Farringdon Lane
London, EC1R 3AW
UK
<http://www.physoc.org>
<http://journals.physoc.org>

REQUIRED ITEMS:

-Include a Key Points list in the article itself, before the Abstract.

-Author photo and profile. First (or joint first) authors are asked to provide a short biography (no more than 100 words for one author or 150 words in total for joint first authors) and a portrait photograph. These should be uploaded and clearly labelled with the revised version of the manuscript. See Information for Authors for further details.

-You must start the Methods section with a paragraph headed Ethical Approval. A detailed explanation of journal policy and regulations on animal experimentation is given in Principles and standards for reporting animal experiments in The Journal of Physiology and Experimental Physiology by David Grundy J Physiol, 593: 2547-2549. doi:10.1113/JP270818.). A checklist outlining these requirements and detailing the information that must be provided in the paper can be found at: <https://physoc.onlinelibrary.wiley.com/hub/animal-experiments>. Authors should confirm in their Methods section that their experiments were carried out according to the guidelines laid down by their institution's animal welfare committee, and conform to the principles and regulations as described in the Editorial by Grundy (2015). The Methods section must contain details of the anaesthetic regime: anaesthetic used, dose and route of administration and method of killing the experimental animals.

-Your manuscript must include a complete Additional Information section

-Please upload separate high-quality figure files via the submission form.

-Please ensure that the Article File you upload is a Word file.

-A Statistical Summary Document, summarising the statistics presented in the manuscript, is required upon revision. It must be on the Journal's template, which can be downloaded from the link in the Statistical Summary Document section here: https://jp.msubmit.net/cgi-bin/main.plex?form_type=display_requirements#statistics

-Papers must comply with the Statistics Policy https://jp.msubmit.net/cgi-bin/main.plex?form_type=display_requirements#statistics

In summary:

-If $n \leq 30$, all data points must be plotted in the figure in a way that reveals their range and distribution. A bar graph with data points overlaid, a box and whisker plot or a violin plot (preferably with data points included) are acceptable formats.

-If $n > 30$, then the entire raw dataset must be made available either as supporting information, or hosted on a not-for-profit repository e.g. FigShare, with access details provided in the manuscript.

- n clearly defined (e.g. x cells from y slices in z animals) in the Methods. Authors should be mindful of pseudoreplication.

- All relevant 'n' values must be clearly stated in the main text, figures and tables, and the Statistical Summary Document (required upon revision)
- The most appropriate summary statistic (e.g. mean or median and standard deviation) must be used. Standard Error of the Mean (SEM) alone is not permitted.
- Exact p values must be stated. Authors must not use 'greater than' or 'less than'. Exact p values must be stated to three significant figures even when 'no statistical significance' is claimed.
- Statistics Summary Document completed appropriately upon revision

-A Data Availability Statement is required for all papers reporting original data. This must be in the Additional Information section of the manuscript itself. It must have the paragraph heading "Data Availability Statement". All data supporting the results in the paper must be either: in the paper itself; uploaded as Supporting Information for Online Publication; or archived in an appropriate public repository. The statement needs to describe the availability or the absence of shared data. Authors must include in their Statement: a link to the repository they have used, or a statement that it is available as Supporting Information; reference the data in the appropriate sections(s) of their manuscript; and cite the data they have shared in the References section. Whenever possible the scripts and other artefacts used to generate the analyses presented in the paper should also be publicly archived. If sharing data compromises ethical standards or legal requirements then authors are not expected to share it, but must note this in their Statement. For more information, see our Statistics Policy.

EDITOR COMMENTS

Reviewing Editor:

Comments for Authors to ensure the paper complies with the Statistics Policy:

Line 320: "All data are presented as mean {plus minus} SEM unless otherwise stated." Per Journal policy, standard deviation must be used throughout.

Lines 412-13: "...skewness of the photocurrent accounts for 98% of the skewness of the voltage responses." Please include the statistical analysis supporting this and others conclusion drawn from the data in Fig. 3.

Error estimates are missing from Figs. 4 and 6.

Comments to the Author:

The paper demonstrates asymmetric gain in cones for positive and negative contrasts and indicates that the asymmetry likely arises in the phototransduction cascade. Modeling analysis reveals why the effects of the asymmetry might have been missed in previous studies. Together the data suggest a potential biophysical basis for the psychophysical phenomenon of skewness discrimination. The referees have raised several issues that would need to be carefully addressed in revision.

As a general comment, the text should be carefully revised throughout to ensure that the conclusions regarding the relevance of the data to previous psychophysical findings are consistent with data presented.

- The title of the paper appears to present a conclusion that is not directly supported by the results. Visual discrimination is not measured.

- The conclusion in the Abstract (lines 27-29) stating "Here we demonstrate that the underlying computation starts as early as the cone phototransduction cascade..." should be more circumspect. The mechanisms described are potentially consistent with human psychophysical results, however, the data all pertain to goldfish cones and therefore do not directly support the contention.

- On lines 33-34, does "perception" refer to fish or is this speculation regarding humans?

- On line 573, "Our results suggest that this discrimination starts ...". They are at least consistent with the notion, given that human cones are reported to behave similarly.

- On line 579: "...in full accordance with psychophysical studies...". Does it make sense to claim full accordance of results obtained in goldfish with human psychophysical studies?

The first reviewer's point #2 relating to the chromatic sensitivity of the cones is worth careful attention, particularly given the strong correlations drawn between the data presented and human psychophysics, and the notion that S-cones are slower than L/M-cones in primates.

The first reviewer's point #4 questions the role of voltage-gated channels. Fig. 3D demonstrates that skewness is similar for current and voltage. Were the response amplitudes sufficient to significantly activate voltage-gated channels? Intuitively one might expect an additional non-linearity to influence the skewness of responses. Further discussion, as the referee suggests, would be helpful.

At several places throughout the paper cones are described as "perceiving" a stimulus. Isn't "perception" a property of higher-level processing?

Please revise Fig. 5C so that the relative heights of the bars are proportional to the values being reported, i.e. the y-axis should start at zero. The graph is visually misleading due to truncation of the y-axis.

Line 554 : The formulation, "No difference neither in A nor B" sounds odd to me. Shouldn't the format be "no difference either in A or B", or, "neither A nor B were different"?

Senior Editor:

Comments for Authors to ensure the paper complies with the Statistics Policy:

They need to use SD and improve data point display in figures

REFeree COMMENTS

Referee #1:

This well-written paper that makes a persuasive case that a psychophysical phenomenon known as the Blackshot mechanism originates with properties of phototransduction in cones. The Blackshot mechanism describes a perceptual phenomenon whereby images in which contrasts that are positively skewed (low negative contrasts punctuated by a few high positive contrasts) can be discriminated from those that are negatively skewed. To analyze this mechanism, the investigators employed a combination of analytical modeling and electrophysiological recordings using a tractable preparation of goldfish cones. They show that the photocurrent responses of goldfish cones differ in their responses to skewed visual stimuli, with a more strongly skewed response to negatively skewed stimulus sequences and more weakly skewed response to positively skewed stimuli. This matches the Blackshot mechanism which also shows a stronger response to negatively skewed contrasts. The study was carefully conceived and well-written. I offer a few suggestions for clarification.

1) The NTSCI stimulus set is a time series of chromatic intensities and the authors use a 3-wavelength LED stimulator. Please provide more detail on the wavelengths used for these experiments and how they might have influenced the analysis. For example, the derivation of "effective" stimuli would vary with the spectral content of the stimulus set and cone subtype (e.g., a large increment in blue would have only small effects on a red-sensitive cone).

2) A related question to point 1: What types of cones did the authors record from? Was it only a single type? If it was more than one subtype, did responses to these stimuli vary among cone subtypes?

3) For the highest skewness of 2.2 in the bar graph of Fig. 5C, only the voltage response ratio was plotted. What was the corresponding current ratio? Was the ratio the same or different?

4) Related to point 3: I would expect that the voltage response to large increments in intensity to be "clipped" by the activation of Ih channels at more negative potentials in the cone. This could contribute to the large increase in the skewness ratio in Fig. 5C at +/-2.2. This reader (and I'm sure others) would benefit from a short discussion about how the ion channels in cones influence the conversion of skewed currents to photovoltage. For modest contrasts, the cone I/V relationship would remain in a pretty linear range, but large increments in intensity can activate Ih and large decrements can begin to activate outward rectifying K+ channels.

I noticed a missing quote mark before "effective" on line 226.

Referee #2:

The manuscript investigates the physiological underpinnings of the perceptual phenomenon of skewness discrimination via current- and voltage-clamp recordings from goldfish cones and model simulations of cone transduction in goldfish and other vertebrates. The manuscript is hard to follow at times but presents a convincing case that the phenomenon can be attributed to cone transduction nonlinearities. The finding is not surprising given known psychophysical and physiological asymmetries in gain to luminance increments and decrements (e.g. Snippe et al. 2004; van Hateren 2005; Endeman & Kamermans, 2010; Angueyra & Rieke, 2013; Angueyra et al. 2021). Response linearity is ultimately an approximation that is valid for a limited contrast range. There would appear little motivation for the work were it not for the more surprising results of two studies of retinal ganglion cells that reported no difference in responses associated with stimulus skewness despite gain asymmetries that have been noted in the retina, LGN, and visual cortex (e.g. Zaghoul et al. 2003; Burkhardt et al. 2004; Freeman et al. 2010; Kremkow et al. 2014). The manuscript not only provides experimental data to support asymmetric cone gain but also nicely shows with model simulations that the white noise stimulus used by those studies contains wasted energy that likely rendered the stimulus ineffective for probing skewness effects. The discussion is vision-centric, limiting the broader potential impact of the result. Many non-vision studies also apply white noise approaches (e.g. van Dijk et al 1997; Massot et al. 2012; Metzen & Chacron 2015) so upstream nonlinearities and "effective" stimulus energy would be worthwhile to consider in the analysis of downstream neural responses more generally. The specific impact of asymmetric photoreceptor gain on vision appears to be minor judging from the subtlety of the skewness effect in Fig.1 (at least to my eyes) and from observations that humans aesthetically prefer non-skewed images (Graham et al. 2016) even though natural images are positively skewed and must be deskewed by cone asymmetries (Figure 8A) based on efficient coding theories.

Additional comments and suggested edits:

L84-97. Much of the text is results and discussion and does not belong in an introduction.

L219-221. Were the pink or skewed segments used to estimate the impulse response function?

L227-30. It seems like the latter from above and the power spectrum of skewed segments is not shown but is presumably non-uniform, which can bias the impulse response estimate. For example, if the stimulus has lower power at a given frequency then response gain would be poorly estimated at that frequency (unless gain is high) and could alter "effective" skewness independent of cone nonlinearity. Why not use the pink segment for all (esp. since it looks noticeably different from the skewed segments in Figure 4A)?

L244-247. Possible bias associated with skewed stimulus is recognized here and now used to justify mean impulse response function instead. This is confusing.

L261-262. Are sig figs correct for all parameters?

L295-299. For purpose of comparison Figure 2 should show recorded impulse response as well.

L368. Panels B and C do not appear to show cone response being more negatively skewed than the stimulus.

L416-421. Convolution of the stimulus with the impulse response would be the "linear predicted response". Calling it an "effective" stimulus perceived by cones obfuscated the question of whether skewness was altered by linear filtering. What is doing the "temporal filtering" if not the cone itself. Implicit here seems to be an idea that there is a cone nonlinearity after the cone "perceives" (aka. filters)?

L431-433. Is the skew stimulus presented raw or filtered? The raw should produce the same skew after cone filtering according to Figure 4B if it is truly an equally "effective" stimulus but I suspect the latter.

L433. No Figure 4C.

L445-447. Is photocurrent or voltage plotted, or both? If latter, why are they not inverted like in 3B and C?

L433-436. Unclear how much the result depends on the one positive and negative segment used, esp. for the negative segment as a cluster of cells fall on the line. Might be a dependence on cone type?

END OF COMMENTS

Confidential Review

12-Jul-2021

EDITOR COMMENTS

Reviewing Editor:

Comments for Authors to ensure the paper complies with the Statistics Policy:

Line 320: "All data are presented as mean {plus minus} SEM unless otherwise stated." Per Journal policy, standard deviation must be used throughout.

We revised manuscript such that it is now fits journal statistical policy.

Lines 412-13: "...skewness of the photocurrent accounts for 98% of the skewness of the voltage responses." Please include the statistical analysis supporting this and others conclusion drawn from the data in Fig. 3.

We made necessary changes in the legend of the Figure 3 and in the parts of the text related to this figure.

Error estimates are missing from Figs. 4 and 6.

We made necessary changes in the legends of the Figures 4&6 and in the corresponding parts of the text.

Comments to the Author:

The paper demonstrates asymmetric gain in cones for positive and negative contrasts and indicates that the asymmetry likely arises in the phototransduction cascade. Modeling analysis reveals why the effects of the asymmetry might have been missed in previous studies. Together the data suggest a potential biophysical basis for the psychophysical phenomenon of skewness discrimination. The referees have raised several issues that would need to be carefully addressed in revision.

As a general comment, the text should be carefully revised throughout to ensure that the conclusions regarding the relevance of the data to previous psychophysical findings are consistent with data presented.

- The title of the paper appears to present a conclusion that is not directly supported by the results. Visual discrimination is not measured.

We changed the title of the paper such that it is now directly supported by the presented results.

- The conclusion in the Abstract (lines 27-29) stating "Here we demonstrate that the underlying computation starts as early as the cone phototransduction cascade..." should be more circumspect.

The mechanisms described are potentially consistent with human psychophysical results, however, the data all pertain to goldfish cones and therefore do not directly support the contention.

We made requested changes in the abstract of the document.

- On lines 33-34, does "perception" refer to fish or is this speculation regarding humans?

We changed abstract to disentangle speculations from the data.

- On line 573, "Our results suggest that this discrimination starts ...". They are at least consistent with the notion, given that human cones are reported to behave similarly.

We changed the discussion to avoid any overstatements.

- On line 579: "...in full accordance with psychophysical studies...". Does it make sense to claim full accordance of results obtained in goldfish with human psychophysical studies?

We changed the discussion to avoid any possible confusions and/or overstatements.

The first reviewer's point #2 relating to the chromatic sensitivity of the cones is worth careful attention, particularly given the strong correlations drawn between the data presented and human psychophysics, and the notion that S-cones are slower than L/M-cones in primates.

We provided more careful description of how employed stimuli were generated to avoid any confusion about their chromatic composition.

The first reviewer's point #4 questions the role of voltage-gated channels. Fig. 3D demonstrates that skewness is similar for current and voltage. Were the response amplitudes sufficient to significantly activate voltage-gated channels? Intuitively one might expect an additional non-linearity to influence the skewness of responses. Further discussion, as the referee suggests, would be helpful.

We added paragraph, where we discuss this issue.

At several places throughout the paper cones are described as "perceiving" a stimulus. Isn't "perception" a property of higher-level processing?

The manuscript was re-written to avoid conflation between low- and high-level visual processing domains.

Please revise Fig. 5C so that the relative heights of the bars are proportional to the values being reported, i.e. the y-axis should start at zero. The graph is visually misleading due to truncation of the y-axis.

The requested changes to the Figure 5C were made.

Senior Editor:

Comments for Authors to ensure the paper complies with the Statistics Policy:

They need to use SD and improve data point display in figures

We changed the manuscript such that now it adheres to the journal data policy.

REFeree COMMENTS

Referee #1:

This well-written paper that makes a persuasive case that a psychophysical phenomenon known as the Blackshot mechanism originates with properties of phototransduction in cones. The Blackshot mechanism describes a perceptual phenomenon whereby images in which contrasts that are positively skewed (low negative contrasts punctuated by a few high positive contrasts) can be discriminated from those that are negatively skewed. To analyze this mechanism, the investigators employed a combination of analytical modeling and electrophysiological recordings using a tractable preparation of goldfish cones. They show that the photocurrent responses of goldfish cones differ in their responses to skewed visual stimuli, with a more strongly skewed response to negatively skewed stimulus sequences and more weakly skewed response to positively skewed stimuli. This matches the Blackshot mechanism which also shows a stronger response to negatively skewed contrasts. The study was carefully conceived and well-written. I offer a few suggestions for clarification.

1) The NTSCI stimulus set is a time series of chromatic intensities and the authors use a 3-wavelength LED stimulator. Please provide more detail on the wavelengths used for these experiments and how they might have influenced the analysis. For example, the derivation of "effective" stimuli would vary with the spectral content of the stimulus set and cone subtype (e.g., a large increment in blue would have only small effects on a red-sensitive cone).

We added more detailed description of the stimuli generation to the methods section such that now the spectral composition of the employed stimuli is clear and unambiguous.

2) A related question to point 1: What types of cones did the authors record from? Was it only a single type? If it was more than one subtype, did responses to these stimuli vary among cone subtypes?

Although majority of the recorded light responses were from M-cones, we recorded from all of the cell types. Responses to the skewed stimuli does not seem to vary with the cell type. However, we must to acknowledge that low number of the recorded S- and L-cones does not allow us to perform meaningful statistical analysis to support this claim.

We also listed number of the recorded cones in the manuscript.

3) For the highest skewness of 2.2 in the bar graph of Fig. 5C, only the voltage response ratio was plotted. What was the corresponding current ratio? Was the ratio the same or different?

The photocurrent responses to the stimuli with the highest skewness were not recorded, since the Figure 3D indicated that the light/dark asymmetry originates in the phototransduction cascade. It is also stated in the text in the subsection about responses to “effective” skew stimuli.

4) Related to point 3: I would expect that the voltage response to large increments in intensity to be "clipped" by the activation of I_h channels at more negative potentials in the cone. This could contribute to the large increase in the skewness ratio in Fig. 5C at ± 2.2 . This reader (and I'm sure others) would benefit from a short discussion about how the ion channels in cones influence the conversion of skewed currents to photovoltage. For modest contrasts, the cone I/V relationship would remain in a pretty linear range, but large increments in intensity can activate I_h and large decrements can begin to activate outward rectifying K^+ channels.

This valid concern. However, in our recording conditions it was not possible to observe the mentioned effect. We added paragraph with the discussion of why it happened this way to the text of the manuscript.

Referee #2:

The manuscript investigates the physiological underpinnings of the perceptual phenomenon of skewness discrimination via current- and voltage-clamp recordings from goldfish cones and model simulations of cone transduction in goldfish and other vertebrates. The manuscript is hard to follow at times but presents a convincing case that the phenomenon can be attributed to cone transduction nonlinearities. The finding is not surprising given known psychophysical and physiological asymmetries in gain to luminance increments and decrements (e.g. Snippe et al. 2004; van Hateren 2005; Endeman & Kamermans, 2010; Angueyra & Rieke, 2013; Angueyra et al. 2021). Response linearity is ultimately an approximation that is valid for a limited contrast range. There would appear little motivation for the work were it not for the more surprising results of two studies of retinal ganglion cells that reported no difference in responses associated with stimulus skewness despite gain asymmetries that have been noted in the retina, LGN, and visual cortex (e.g. Zaghoul et al. 2003; Burkhardt et al. 2004; Freeman et al. 2010; Kremkow et al. 2014). The manuscript not only provides experimental data to support asymmetric cone gain but also nicely shows with model simulations that the white noise stimulus used by those studies contains wasted energy that likely rendered the stimulus ineffective for probing skewness effects. The discussion is vision-centric, limiting the broader potential impact of the result. Many non-vision studies also apply white noise approaches (e.g. van Dijk et al 1997; Massot et al. 2012; Metzen & Chacron 2015) so upstream nonlinearities and "effective" stimulus energy would be worthwhile to consider in the analysis of downstream neural responses more generally. The specific impact of asymmetric photoreceptor gain on vision appears to be minor judging from the subtlety of the skewness effect in Fig.1 (at least to my eyes) and from observations that humans aesthetically prefer non-skewed images (Graham et al. 2016) even though natural images are positively skewed and must be deskewed by cone asymmetries (Figure 8A) based on efficient coding theories.

Additional comments and suggested edits:

L84-97. Much of the text is results and discussion and does not belong in an introduction.

We thank reviewer for the suggestion. We really tried to shorten it, but found it to be hardly possible. Additionally, we feel like it is important to brief reader about what we found and what does it mean in the introduction, because it immediately provides reader with the key information about presented data.

L219-221. Were the pink or skewed segments used to estimate the impulse response function?

We made some changes in our estimates of the impulse response functions. To estimate "effective" skewed stimuli impulse response functions were calculated from the entire Skew Stimulus Set stretch to avoid any biases linked to the lack of symmetry in the skewed stimuli.

L227-30. It seems like the latter from above and the power spectrum of skewed segments is not shown but is presumably non-uniform, which can bias the impulse response estimate. For example, if the stimulus has lower power at a given frequency then response gain would be poorly estimated at that frequency (unless gain is high) and could alter "effective" skewness independent of cone nonlinearity. Why not use the pink segment for all (esp. since it looks noticeably different from the skewed segments in Figure 4A)?

As it is stated above, we changed the way of the estimation of the impulse response functions for the calculation of the "effective" skewed. Now we calculated them from the entire stretches of the Stimuli Stretches.

L244-247. Possible bias associated with skewed stimulus is recognized here and now used to justify mean impulse response function instead. This is confusing.

The biases associated with skewness does not allow to estimate "true" impulse response function and obscure estimates of the filter gain. Therefore, to avoid such biases and obtain "true" impulse response function one should estimate it from the entire Skew Stimulus Set, which is elliptically symmetrical. Usually, during calculation of the impulse response functions people use windows to split the data into stretches and average those estimates. Therefore, given linearity of these steps, it is essentially the same as to calculate impulse response functions for each skew trace and then average obtain impulse response functions.

The biases associated with the skewness does not allow to estimate "true" impulse response function. Yet, since the only difference between positively and negatively skewed stimuli is their skewness, one can still infer differential effect of the stimulus skewness on the cone processing by comparing impulse response functions obtained from the responses to positively and negatively skewed stimuli.

L261-262. Are sig figs correct for all parameters?

Yes, the parameters are correct I think the reviewer might be concerned that that if one will put into equation contrast of zero, you will get the voltage response of -0.06mV instead of

0mV. However, to get zero one should change value of the parameter b only by 0.011, which well within estimated 95% confidence interval of 0.072.

L295-299. For purpose of comparison Figure 2 should show recorded impulse response as well.

We thank reviewer for the suggestion. However, the efficacy of the model on the Figure 2 is already shown by the R^2 between traces. The goal of the shown impulse response is to indicate that impulse response functions estimated from simulated responses to negatively and positively skewed stimuli retain key characteristics of those estimated from the recorded impulse response functions. This is further reinforced in the Figure legend and in the materials and methods section related to the "Model", where we show that the differences in the impulse response functions calculated from the simulated responses. What is crucial there is not the exact correspondence between recorded and simulated impulse response, but the fact that model captures differential nature of the stimulus skewness on the photoreceptor response dynamics.

L368. Panels B and C do not appear to show cone response being more negatively skewed than the stimulus.

We thank reviewer for the suggestion. We rephrased the description of the corresponding panels.

L416-421. Convolving the stimulus with the impulse response would be the "linear predicted response". Calling it an "effective" stimulus perceived by cones obfuscated the question of whether skewness was altered by linear filtering. What is doing the "temporal filtering" if not the cone itself. Implicit here seems to be an idea that there is a cone nonlinearity after the cone "perceives" (aka. filters)?

We rewrite manuscript to avoid any "perception by cone photoreceptors". We also add to the text justification of the word "effective" and why it serves the purpose of the presentation of our results better than "linear prediction"

L431-433. Is the skew stimulus presented raw or filtered? The raw should produce the same skew after cone filtering according to Figure 4B if it is truly an equally "effective" stimulus but I suspect the latter.

We selected stimuli based on whether their skewness is affected by the temporal filtering. We found subset of stimuli for which skewness remains unaffected by the convolution with mean filter (albeit, it slightly affects skewness of the zero skewed stimulus). Then this raw stimulus was presented and voltage responses of the cells was recorded.

L433. No Figure 4C.

Fixed.

L445-447. Is photocurrent or voltage plotted, or both? If latter, why are they not inverted like in 3B and C?

Both. Actually, they are inverted. As otherwise, dots for voltage responses would be positive for negatively skewed stimuli.

L433-436. Unclear how much the result depends on the one positive and negative segment used, esp. for the negative segment as a cluster of cells fall on the line. Might be a dependence on cone type?

The real reason behind is the presence or absence of strong negative contrasts. As this is the major factor dictating deviations between skews of the stimulus and response.

Dear Professor Kamermans,

Re: JP-RP-2021-282152R1 "Enhancing the dark side: Asymmetric gain of cone photoreceptors underpins their discrimination of visual scenes based on their skewness" by Matthew Yedutenko, Marcus H.C. Howlett, and Maarten Kamermans

Thank you for submitting your revised Research Paper to The Journal of Physiology. It has been assessed by the original Reviewing Editor and Referees and has been well received. Some final revisions have been requested.

The reports are copied at the end of this email. Please address all of the points and incorporate all requested revisions, or explain in your Response to Referees why a change has not been made.

NEW POLICY: In order to improve the transparency of its peer review process The Journal of Physiology publishes online as supporting information the peer review history of all articles accepted for publication. Readers will have access to decision letters, including all Editors' comments and referee reports, for each version of the manuscript and any author responses to peer review comments. Referees can decide whether or not they wish to be named on the peer review history document.

Authors are asked to use The Journal's premium BioRender (<https://biorender.com/>) account to create/redrawn their Abstract Figures. Information on how to access The Journal's premium BioRender account is here: <https://physoc.onlinelibrary.wiley.com/journal/14697793/biorender-access> and authors are expected to use this service. This will enable Authors to download high-resolution versions of their figures.

I hope you will find the comments helpful and have no difficulty returning your revisions within two weeks.

Your revised manuscript should be submitted online using the links in Author Tasks Link Not Available.

Any image files uploaded with the previous version are retained on the system. Please ensure you replace or remove all files that have been revised.

REVISION CHECKLIST:

- Article file, including any tables and figure legends, must be in an editable format (eg Word)
- Abstract figure file (see above)
- Statistical Summary Document
- Upload each figure as a separate high quality file
- Upload a full Response to Referees, including a response to any Senior and Reviewing Editor Comments;
- Upload a copy of the manuscript with the changes highlighted.

- A potential 'Cover Art' file for consideration as the Issue's cover image;
- Appropriate Supporting Information (Video, audio or data set https://jp.msubmit.net/cgi-bin/main.plex?form_type=display_requirements#supp).

To create your 'Response to Referees' copy all the reports, including any comments from the Senior and Reviewing Editors, into a Word, or similar, file and respond to each point in colour or CAPITALS and upload this when you submit your revision.

I look forward to receiving your revised submission.

If you have any queries please reply to this email and staff will be happy to assist.

Yours sincerely,

Ian D. Forsythe

Deputy Editor-in-Chief
The Journal of Physiology
<https://jp.msubmit.net>
<http://jp.physoc.org>
The Physiological Society
Hodgkin Huxley House
30 Farringdon Lane
London, EC1R 3AW
UK
<http://www.physoc.org>
<http://journals.physoc.org>

REQUIRED ITEMS:

-Author photo and profile. First (or joint first) authors are asked to provide a short biography (no more than 100 words for one author or 150 words in total for joint first authors) and a portrait photograph. These should be uploaded and clearly labelled with the revised version of the manuscript. See Information for Authors for further details.

-You must start the Methods section with a paragraph headed Ethical Approval. A detailed explanation of journal policy and regulations on animal experimentation is given in Principles and standards for reporting animal experiments in The Journal of Physiology and Experimental Physiology by David Grundy J Physiol, 593: 2547-2549. doi:10.1113/JP270818.). A checklist outlining these requirements and detailing the information that must be provided in the paper can be found at: <https://physoc.onlinelibrary.wiley.com/hub/animal-experiments>. Authors should confirm in their Methods section that their experiments were carried out according to the guidelines laid down by their institution's animal welfare committee, and conform to the principles and regulations as described in the Editorial by Grundy (2015). The Methods section must contain details of the anaesthetic regime: anaesthetic used, dose and route of administration and method of killing the experimental animals.

-Papers must comply with the Statistics Policy https://jp.msubmit.net/cgi-bin/main.plex?form_type=display_requirements#statistics

In summary:

-If $n \leq 30$, all data points must be plotted in the figure in a way that reveals their range and distribution. A bar graph with data points overlaid, a box and whisker plot or a violin plot (preferably with data points included) are acceptable formats.

-If $n > 30$, then the entire raw dataset must be made available either as supporting information, or hosted on a not-for-profit repository e.g. FigShare, with access details provided in the manuscript.

- n clearly defined (e.g. x cells from y slices in z animals) in the Methods. Authors should be mindful of pseudoreplication.

-All relevant n values must be clearly stated in the main text, figures and tables, and the Statistical Summary Document (required upon revision)

-The most appropriate summary statistic (e.g. mean or median and standard deviation) must be used. Standard Error of the Mean (SEM) alone is not permitted.

-Exact p values must be stated. Authors must not use 'greater than' or 'less than'. Exact p values must be stated to three significant figures even when 'no statistical significance' is claimed.

-Statistics Summary Document completed appropriately upon revision

EDITOR COMMENTS

Reviewing Editor:

There are a few minor issues that need to be addressed as noted by the second referee.

Animals access to food and water: the latter is clear enough but the former is not mentioned.

I agree with referee #2 regarding the Introduction. Up until line 103, the introduction provides a very nice summary of the background and states the main goal of the study, however, the text on lines 103-116 starting with "To account for the kinetics of cone photoreceptors..." includes methods, results, discussion, and conclusions that would be better moved to the relevant sections, or deleted if repetitious.

Senior Editor:

Comments for Authors to ensure the paper complies with the Statistics Policy:

There is one bar graph figure 5C that requires the raw data points.

The revision is satisfactory and there are just a few minor points to cover before final acceptance.

REFEREE COMMENTS

Referee #1:

My earlier concerns and questions have all been addressed. I have no further concerns.

Referee #2:

see attached.

END OF COMMENTS

1st Confidential Review

21-Sep-2021

The manuscript has been substantially revised with expanded and improved descriptions of the light stimulus and data analysis methods. The readability is greatly improved.

Some additional comments:

L107-116. Again, text is unnecessary for introduction. Nevertheless, the authors believe it is important to foreshadow the main finding. L104-106 does this. The rest of the paragraph can be deleted without loss to reader as the specific results and their meaning are provided in the abstract and are detailed in results and discussion.

L487-489. It is still not clear why a white or pink noise stimulus was not used to estimate the impulse response (data were collected for pink noise). While the entire Skew Stimulus Set#1 may have zero “net skew” (elliptical intensity symmetry) based on its construction, it is still not temporally random and might not therefore drive cones adequately at some frequencies to evoke measurable responses at those frequencies (e.g. contrast energy declines with stimulus frequency which may lead to lower “effective” cone cutoff frequency).

Fig 5. Panel B has “effective” stimulus skews in the range of -3 to -4, yet text states that only ± 0.1 , ± 0.6 , ± 1.6 , and ± 2.2 were presented. Legend says that cone responses were “more skewed when high levels of negative effective skews are delivered”, but cone skews were around -2 to -2.5 which is less not more. Are the axes backwards?

Line 637. Typo, should be “assess”.

Line 706. Delete “the”.

Line 736. Typo, should be “affected”.

Responses to comments

Reviewing Editor

Comment: Animals access to food and water: the latter is clear enough but the former is not mentioned.

Response: We made necessary changes into the text

Comment: I agree with referee #2 regarding the Introduction. Up until line 103, the introduction provides a very nice summary of the background and states the main goal of the study, however, the text on lines 103-116 starting with "To account for the kinetics of cone photoreceptors..." includes methods, results, discussion, and conclusions that would be better moved to the relevant sections, or deleted if repetitious.

Response: We thank editor for the comment; we made corresponding changes in the text.

Comments for Authors to ensure the paper complies with the Statistics Policy

Comment: There is one bar graph figure 5C that requires the raw data points.

Response: We made necessary changes to the Figure 5C

Reviewer#2

Comment: L107-116. Again, text is unnecessary for introduction. Nevertheless, the authors believe it is important to foreshadow the main finding. L104-106 does this. The rest of the paragraph can be deleted without loss to reader as the specific results and their meaning are provided in the abstract and are detailed in results and discussion.

Response: We thank reviewer for the comment; we made corresponding changes in the text.

Comment: L487-489. It is still not clear why a white or pink noise stimulus was not used to estimate the impulse response (data were collected for pink noise). While the entire Skew Stimulus Set#1 may have zero "net skew" (elliptical intensity symmetry) based on its construction, it is still not temporally random and might not therefore drive cones adequately at some frequencies to evoke measurable responses at those frequencies (e.g. contrast energy declines with stimulus frequency which may lead to lower "effective" cone cutoff frequency).

Response:

We thank the reviewer for their comment and recognize their point regarding the use of spherically symmetrical stimulus (like white noise) to estimate “true” cellular impulse response functions. However, at the photopic light intensities we used it is generally not possible to maintain stable light responses long enough to collect data for both white noise and skewed stimuli. We suggest our approach was justifiable, and any potential shortcoming had little to no effect on the outcomes or conclusion of this paper, for the following reasons:

1) The focus of the present paper is the differential responses of the cone photoreceptors to positive and negative contrast. Linear filtering estimates of ‘effective’ skew levels, regardless of how precisely it replicates the ‘true’ linear filtering employed by the cones, does not change the observation that cone responses differed under positive and negative skew conditions.

2) The use of natural stimuli could underestimated the cone cut-off frequencies, and this may well cause issues if we were to draw general conclusions regarding cone kinetic responses to any arbitrary stimuli. However, here this should have little effect on our estimates of ‘effective’ skew as, as the reviewer rightly points out, the stimuli themselves have very little power at higher frequencies. Hence, higher frequency contributions to the overall skewness of the stimulus are minimal and as such any over-filtering will have little effect on the outcome.

3) To reassure the reviewer and ourselves that using the skew-stimuli derived impulse functions did not lead to grossly inaccurate estimates of ‘effective’ skew levels, we determined what the skew levels were when using an impulse response function from one of our previous publication (Howlett et al, PLoS Biol, 2017). In this instance the impulse function was derived from M and L cone voltage responses to a mixed sine stimulus consisting of 21 equal amplitude sine waves ranging in frequency from 0.3Hz to 31.7 Hz. As the figures below show, there is very little difference between the ‘effective’ skew values shown in the current manuscript (black squares) and those derived using the mixed sine stimulus (red circles)

Comment: Fig 5. Panel B has “effective” stimulus skews in the range of -3 to -4, yet text states that only ± 0.1 , ± 0.6 , ± 1.6 , and ± 2.2 were presented. Legends says that cone responses were “more skewed when high levels of negative effective skews are delivered”, but cone skews were around -2 to -2.5 which is less not more. Are the axes backwards?

Response: The X axis correspond to the “Effective” skew values, the Y axis corresponds to the responses skew values. As it shown on the Figure 5B, on Y axis data points are between -4 and 2. The reason why we also have X axis from -4 and 4 is that we wanted our plots to be of the square shape. We did so, because in our opinion such a layout will make differences between stimulus and response skews more vivid.

Comment: Line 637. Typo, should be “assess”.

Response: Fixed

Comment: Line 706. Delete “the”.

Response: Fixed

Comment: Line 736. Typo, should be “affected”.

Response: Fixed

Finally, we made a small correction in Fig. 1 and the associated legend.

Dear Professor Kamermans,

Re: JP-RP-2021-282152R2 "Enhancing the dark side: Asymmetric gain of cone photoreceptors underpins their discrimination of visual scenes based on their skewness" by Matthew Yedutenko, Marcus H.C. Howlett, and Maarten Kamermans

I am pleased to tell you that your paper has been accepted for publication in The Journal of Physiology, subject to any modifications to the text and/or satisfactory clarification of the Methods section that may be required by the Journal Office to conform to House rules.

NEW POLICY: In order to improve the transparency of its peer review process The Journal of Physiology publishes online as supporting information the peer review history of all articles accepted for publication. Readers will have access to decision letters, including all Editors' comments and referee reports, for each version of the manuscript and any author responses to peer review comments. Referees can decide whether or not they wish to be named on the peer review history document.

The last Word version of the paper submitted will be used by the Production Editors to prepare your proof. When this is ready you will receive an email containing a link to Wiley's Online Proofing System. The proof should be checked and corrected as quickly as possible.

Authors should note that it is too late at this point to offer corrections prior to proofing. Major corrections at proof stage, such as changes to figures, will be referred to the Reviewing Editor for approval before they can be incorporated. Only minor changes, such as to style and consistency, should be made a proof stage. Changes that need to be made after proof stage will usually require a formal correction notice.

All queries at proof stage should be sent to TJP@wiley.com

The accepted version of the manuscript will be published online, prior to copy editing, in the Accepted Articles section.

Are you on Twitter? Once your paper is online, why not share your achievement with your followers. Please tag The Journal (@jphysiol) in any tweets and we will share your accepted paper with our 22,000+ followers!

Yours sincerely,

Ian D. Forsythe
Deputy Editor-in-Chief
The Journal of Physiology
<https://jp.msubmit.net>
<http://jp.physoc.org>
The Physiological Society
Hodgkin Huxley House
30 Farringdon Lane
London, EC1R 3AW
UK
<http://www.physoc.org>
<http://journals.physoc.org>

P.S. - You can help your research get the attention it deserves! Check out Wiley's free Promotion Guide for best-practice recommendations for promoting your work at www.wileyauthors.com/eoo/guide. And learn more about Wiley Editing Services which offers professional video, design, and writing services to create shareable video abstracts, infographics, conference posters, lay summaries, and research news stories for your research at www.wileyauthors.com/eoo/promotion.

* IMPORTANT NOTICE ABOUT OPEN ACCESS *

Information about Open Access policies can be found here <https://physoc.onlinelibrary.wiley.com/hub/access-policies>

To assist authors whose funding agencies mandate public access to published research findings sooner than 12 months after publication The Journal of Physiology allows authors to pay an open access (OA) fee to have their papers made freely available immediately on publication.

You will receive an email from Wiley with details on how to register or log-in to Wiley Authors Services where you will be able to place an OnlineOpen order.

You can check if your funder or institution has a Wiley Open Access Account here <https://authorservices.wiley.com/author-resources/Journal-Authors/licensing-and-open-access/open-access/author-compliance-tool.html>

Your article will be made Open Access upon publication, or as soon as payment is received.

If you wish to put your paper on an OA website such as PMC or UKPMC or your institutional repository within 12 months of publication you must pay the open access fee, which covers the cost of publication.

OnlineOpen articles are deposited in PubMed Central (PMC) and PMC mirror sites. Authors of OnlineOpen articles are permitted to post the final, published PDF of their article on a website, institutional repository, or other free public server, immediately on publication.

Note to NIH-funded authors: The Journal of Physiology is published on PMC 12 months after publication, NIH-funded authors DO NOT NEED to pay to publish and DO NOT NEED to post their accepted papers on PMC.

EDITOR COMMENTS

Reviewing Editor:

Very nice paper. No further comments.

2nd Confidential Review

26-Oct-2021